

# REVIEW ARTICLE: Drought as a continuum: memory effects in interlinked hydrological, ecological, and social systems

Anne F. Van Loon[1], Sarra Kchouk[2,*], Alessia Matanó[1,*], Faranak Tootoonchi[3], Camila Alvarez-Garreton[4], Khalid E.A. Hassaballah[5], Minchao Wu[6,7], Marthe L.K. Wens[1], Anastasiya Shyrokaya[8], Elena Ridolfi[9], Riccardo Biella[8], Viorica Nagavciuc[10,11], Marlies H. Barendrecht[12,1], Ana Bastos[13], Louise Cavalcante[14], Franciska T. de Vries[15], Margaret Garcia[16,17], Johanna Mård[8], Ileen N. Streefkerk[1], Claudia Teutschbein[6], Roshanak Tootoonchi[18], Ruben Weesie[1], Valentin Aich[19], Juan P. Boisier[4,20], Giuliano Di Baldassarre[8], Yiheng Du[21], Mauricio Galleguillos[4,22], René Garreaud[4,20], Monica Ionita[13], Sina Khatami[23], Johanna K.L. Koehler[14,1,24], Charles H. Luce[25], Shreedhar Maskey[26], Heidi D. Mendoza[1], Moses N. Mwangi[27], Ilias G. Pechlivanidis[21], Germano G. Ribeiro Neto[28], Tirthankar Roy[29], Robert Stefanski[30], Patricia Trambauer[31], Elizabeth A. Koebele[32,33], Giulia Vico[34], Micha Werner[26]

[1] Institute for Environmental Studies (IVM), Vrije Universiteit Amsterdam, 1081 HV Amsterdam, the Netherlands
[2] Water Resources Management Group, Wageningen University, the Netherlands
[3] Department of crop production ecology, Swedish university of agricultural science (SLU), Uppsala, Sweden
[4] Center for Climate and Resilience Research (ANID/FONDAP/1522A0001), Santiago, Chile
[5] IGAD Cilmate Prediction and Applications Center, Nairobi, Kenya
[6] Department of Earth Sciences, Uppsala University, Sweden
[7] Department of Physical Geography and Ecosystem Science, Lund University, Sweden
[8] Centre of Natural Hazards and Disaster Science (CNDS), Department of Earth Sciences, Uppsala University, Uppsala, Sweden
[9] Dipartimento di Ingegneria Civile, Edile e Ambientale, Università degli Studi di Roma La Sapienza, 00184 Rome, Italy
[10] Alfred Wegener Institute for Polar and Marine Research, Bremerhaven, Germany
[11] Faculty of Forestry, "Ştefan cel Mare" University of Suceava, Universităţii street, no.13, 720229, Suceava, Romania
[12] Department of Geography, King's College London, London, United Kingdom
[13] Max Planck Institute for Biogeochemistry, Dept. of Biogeochemical Integration, Jena, Germany
[14] Public Administration and Policy Group, Wageningen University, The Netherlands
[15] Institute for Biodiversity and Ecosystem dynamics, University of Amsterdam
[16] School of Sustainable Engineering & the Built Environment, Arizona State University
[17] Senior Global Futures Scientist, Julie Ann Wrigley Global Futures Laboratory
[18] Department of Civil, Environmental and Mechanical Engineering, University of Trento, Italy
[19] Global Water Partnership
[20] Department of Geophysics, Universidad de Chile, Santiago, Chile
[21] Hydrology Research Unit, Swedish Meteorological and Hydrological Institute, Norrköping, Sweden
[22] Facultad de Ingeniería y Ciencias, Unversidad Adolfo Ibañez, Santiago, Chile
[23] Department of Infrastructure Engineering, University of Melbourne, Parkville, Australia
[24] School of Geography and the Environment, University of Oxford, UK
[25] USDA Forest Service Research and Development, Boise, Idaho, United States
[26] IHE Delft Institute for Water Education, Department of Water Resources and Ecology, The Netherlands
[27] South Eastern Kenya University
[28] Hydrology and Quantitative Water Management group , Wageningen University, the Netherlands
[29] University of Nebraska-Lincoln, United States
[30] World Meteorological Organization (WMO)
[31] Deltares, The Netherlands



[32] Political Science, University of Nevada-Reno, Reno, United States
      [33] Graduate Program of Hydrologic Sciences, University of Nevada-Reno, Reno, United States
      [34] Department of Ecology, Swedish University of Agricultural Sciences (SLU), Uppsala, Sweden

      * These authors contributed equally to the paper.

*Correspondence to*: Anne F. Van Loon (anne.van.loon@vu.nl)

**Abstract.** Droughts are often long lasting phenomena, without a distinct start or end, and with impacts cascading across sectors and systems, creating long-term legacies. Nevertheless, our current perception and management of droughts and their impacts is often event-based, which can limit the effective assessment of drought risks and reduction of drought impacts. Here, we advocate for changing this perspective and viewing drought as a hydro-eco-social continuum. We take a systems

theory perspective and focus on how "memory" causes feedback and interactions between parts of the interconnected systems at different time scales. We first discuss the characteristics of the drought continuum with a focus on the hydrological, ecological, and social systems separately; and then study the system of systems. Our analysis is based on a review of the literature and a study of five cases: Chile, the Colorado River Basin in the US, Northeast Brazil, Kenya, and the Rhine River Basin in Northwest Europe. We find that the memories of past dry and wet periods, carried by both bio-

physical (e.g. groundwater, vegetation) and social systems (e.g. people, governance), influence how future drought risk manifests. We identify four archetypes of drought dynamics: Impact & recovery; Slow resilience-building; Gradual collapse; and High resilience, big shock. The interactions between the hydrological, ecological and social systems result in systems shifting between these types, which plays out differently in the five case studies. We call for more research on drought pre-conditions and recovery in different systems, on dynamics cascading between systems and triggering system changes, and on

dynamic vulnerability and maladaptation. Additionally, we argue for more continuous monitoring of drought hazards and impacts, modelling tools that better incorporate memories and adaptation responses, and management strategies that increase social and institutional memory to better deal with the complex hydro-eco-social drought continuum and identify effective pathways to adaptation.

## 1 Introduction

Drought is a creeping phenomenon (Wilhite and Glantz, 1985) with unclear definitions of when a dry spell develops into a drought (Hall and Leng, 2019; Slette et al., 2019). This is what we read in the introduction of almost every drought paper and what we have been taught at school or university. However, in drought monitoring, analysis, and management drought is still framed as an event. For example, most disaster databases record only within-year events and do so in a binary way (drought / no drought; e.g. EM-DAT, 2023). Yet, multiple failed rainy seasons cause exponentially more harm than a single failed

season, as for example recently seen in the Horn of Africa (Amha et al., 2023). Also in ecosystems, consecutive droughts cause legacies, affecting these ecosystems' long-term resilience (Müller and Bahn, 2022). In many places, such as the Netherlands, drought monitors and management committees are only operational in the summer period when most impacts





are expected, or are only put in place once a defined drought event is underway (KMNI, 2023). This is problematic because drought impacts in one season or year are strongly dependent on what happened in previous seasons and years (i.e.

antecedent conditions and baseline vulnerability to drought), and on what happens afterwards (i.e. responses to and recovery from drought). The responses of a system to alterations of dry and wet periods are related to the "memory" in the system. Here, we argue that the event-based approach to understanding and managing drought needs to change if we want to better mitigate drought impacts on both ecosystems and society. We make this argument by exploring and discussing memory effects in interacting hydrological, ecological, and social systems based on literature and narratives from five global cases.

Drought has different faces, and complexities are inherent in each aspect of drought, including hazard, impacts and overall risk. Drought hazards can manifest in different parts of the hydrological system, propagating from meteorological drought to soil moisture drought and hydrological drought, with different spatial and temporal characteristics (Van Loon, 2015). There are different temporal dimensions to drought, which range from flash droughts to mega-droughts (Christian et al., 2021; Cook et al., 2022; Ionita et al., 2021; Pendergrass et al., 2020). Flash droughts are driven by a short but extreme precipitation

deficit (often co-occurring with high evapotranspiration rates (Shah et al., 2022)) and mega-droughts by a less extreme, but very prolonged reduction in precipitation and/or increase in evaporative demand (Ault et al., 2016). Not only drought development, but also recovery from drought varies in space and time and between different parts of the hydrological system. This means that the end of a drought is not always clear and dry conditions may linger for a long time (Parry et al., 2016; Tijdeman et al., 2022). Drought hazards are quantified with a range of different indices (such as the Standardised

Precipitation Index, SPI (McKee et al., 1993)), the Standardized Precipitation Evapotranspiration Index (SPEI), (Vicente-Serrano et al., 2010)), the self-calibrated Palmer Drought Severity Index (Wells et al., 2004), among others), many of which allow for considering different levels of severity and different time periods (accumulation of precipitation over several timescales). However, in event-based analyses these are again reduced to a single timescale and a threshold is used to define a distinct start and end date for a specific drought event (Brunner et al., 2021; Kchouk et al., 2022; Van Loon, 2015).

Drought impacts on ecosystems and society are wide-ranging, and extend across a variety of temporal and spatial scales. The delineation of what a drought impact is and when it starts and ends is not straightforward (Hall and Leng, 2019; Slette et al., 2019) and the timescales of drought impacts are highly variable (de Brito et al., 2020). Drought impacts are often gradual in time, for example increased walking distance to collect water due to drying up of boreholes, progressive vegetation stress due to decreasing soil moisture levels, reduced energy production or goods transported due to reduced river water levels,

affected livelihoods leading to school dropouts. Some of these changes are not even reported as impact, but rather as a way of coping with drought. This makes relating drought impacts to drought hazard indicators challenging (Bachmair et al., 2016; Shyrokaya et al., 2024). A way to include social dynamics is to add vulnerability factors (Blauhut et al., 2016). Other studies use continuous impact data and relate these to gradual drought severity levels with continuous indices (e.g. crop yield; (Madadgar et al., 2017)). This approach can capture non-linear relationships, but still does not take into account the fact that

vulnerability can also change during drought, thereby affecting future drought impacts (i.e. vulnerability is dynamic; (de Ruiter and van Loon, 2022)).



Drought risks are complex (Blauhut, 2020). Interactions between hazard, vulnerability and impacts are not always one-way, and feedback, human (re)actions and cascading effects make the relationship between drought hazard and impacts non-linear and dynamic (de Brito, 2021; Kchouk et al., 2022; Wens et al., 2019). For example, when agricultural crops are affected by
soil moisture drought, impacts may initially be mitigated by applying irrigation (e.g. from groundwater), but later this water abstraction enhances hydrological drought, which in turn impacts other sectors dependent on groundwater (e.g. drinking water supply; (Pauloo et al., 2020)). These interactions and non-linearities are often not included in drought hazard-impacts studies (Wens et al., 2019), and studies on future drought risk only consider changes in drought hazard, keeping exposure and vulnerability fixed in time (Hagenlocher et al., 2019). In addition, the perception of drought severity, impacts, planning,
and management can also considerably differ among different societies or communities, as well as over time, and may not always align with the actual observed drought severity and impacts (Teutschbein et al., 2023).

Here, we argue that, taking an event-based view of drought, i.e. monitoring or assessing drought as a snapshot in time (Bastos et al., 2023), and analysing each drought driver or impact separately, oversimplifies the complexity of drought and its dynamics (Hagenlocher et al., 2023). Such an event-based approach overlooks important periods that shape antecedent
conditions and recovery from drought, as well as the cascading and compounding effects of drought processes, dynamic vulnerability, and feedback between hazard and impact. Moreover, because of the complex interactions between drought and multiple systems, we argue that understanding drought requires taking into account not only physical (hydro-meteorological) processes, but also ecological (environmental) and social (economical, political) processes to assess drought risks. There is a strong need to view drought as systemic risk  (Sillmann et al., 2022; Hagenlocher et al., 2023). To achieve these aims, we
use systems theory and link this to the temporal dimensions of drought (Section 2), discuss memory in hydrological, ecological and social systems (Section 3), and compare the emerging properties and feedback in / between these systems (Section 4). We base our analysis on a review of the scientific literature and on five case studies in different parts of the world (see Appendix 1-5). We conclude with an outlook providing recommendations for further research and improved monitoring and management (Section 5). Considering these aspects is needed for a new perspective on drought, wherein we
conceptualize drought as continuous and consisting of interacting hydro-eco-social memory processes.

## 2 Drought in the complex hydro-eco-social system

In this paper we build on the concepts of complex systems and systems thinking to conceptualize drought as a hydro-eco-social system, and draw on elements from social-ecological systems, socio-hydrology, and earth system science. Our specific focus is the dynamic aspects of these interacting systems interacting over time as they are affected by and create system
"memory". In this section, we first introduce key overarching concepts relevant to our conceptualization of the drought system and its temporal dimensions.

The field of systems thinking defines complex systems as composed of a set of elements (which can be systems themselves) that have connections between each other (Jackson, 2019; Shaked and Schechter, 2017). The interactions between these



interconnected elements can lead to unexpected emergent results (Westra and Zscheischler, 2023). Elements can interact and feedback at different scales, creating a multidimensional complex adaptive system (Rammel et al., 2007). Systems theory is, for example, applied to agriculture, natural resources management (Ison et al., 1997), and disaster recovery (Bahmani and Zhang, 2021).

Social-ecological systems (SES) are examples of complex adaptive systems characterised by integrated bio-physical and socio-cultural processes (Ahmed and Abdalla, 2005; Delgado-Serrano et al., 2015; Ostrom, 2009; Tellman et al., 2018). Socio-hydrology or hydrosocial systems can be seen as a specific SES revolving around the interactions between people and water (Konar et al., 2019; Sivapalan et al., 2012; Wesselink et al., 2017). Many studies, for example, use socio-hydrology to understand and model the complex dynamics of flood risk resulting from the interplay between floods and people (Di Baldassarre et al., 2013; Vanelli et al., 2022). Earth system science (ESS) focuses on the complex adaptive components of the earth system and their interactions (Steffen et al., 2020). ESS is strongly based in the natural sciences (meteorology, climate physics, environmental science), but has more recently recognised the important role of humans as agents of change of the earth system (Alessa and Chapin, 2008). One difference between SES and ESS is the scale at which they are studied, with ESS focusing on the planetary scale (Steffen et al., 2020).

Within complex social-ecological or earth systems, the interactions between the elements or subsystems happen across both spatial and temporal scales (Konar et al., 2019). In this paper, we are interested in temporal aspects. Naylor et al., (2020) state that to understand complex systems and their emergent properties, it is necessary to examine the changes in relationships between system elements over time. The concept of time is studied extensively in the separate systems - the hydrological system (Koutsoyiannis, 2013), ecosystem (Jackson et al., 2021), and social system (Peixoto and Rosvall, 2017) - despite common features between them. Aspects like antecedent conditions, response times to a disturbance, and recovery to the original state (or transition to a new state) jointly shape the response of a system to external drivers, resulting in a quickly or slowly changing system. This temporal system response is conceptualised as the "memory" of a system, with a short system memory leading to a quick response and short legacies, and a long system memory leading to a slow response and long legacies (Gunderson and Holling, 2002; Kchouk et al., 2023; Redman and Kinzig, 2003).

The memory of the systems within a complex system strongly determines the emerging properties, such as: i) self-organization and emergence, ii) non-linear behaviour and tipping points, iii) state shifts and feedback loops, and iv) resilience and adaptation (Carmichael and Hadžikadić, 2019; Preiser et al., 2018). Such properties are particularly evident when examining the co-evolution of human and water systems across time. For example, Srinivasan et al., (2012) introduced the concept of 'syndromes' to conceptualise and describe the evolving nature of human-water interactions over time. These 'syndromes' represent specific patterns of water use, reflecting the dynamic state of the system as it changes and adapts with time. Similarly, Roobavannan et al., (2017) modelled a 'pendulum swing' in the management of the Murrumbidgee Basin in Australia, which is in fact a shift from agricultural to environmental water allocation. This shift reflects the 'memory properties' of systems as it was shaped by accumulated experiences, past policies, and societal values, showing how historical experiences influence current practices.





Time is an important element in the development of drought and drought impacts, as recognised by previous studies (Hall and Leng, 2019; Tijdeman et al., 2022; Wilhite and Glantz, 1985; WMO, 2021) and time characteristics have been studied
empirically in the separate systems (see some examples in Table 1). In the next sections, we explore and discuss the concept of memory shaping drought over time from different perspectives: hydrology, ecology, and social science. Next, we analyze potential temporal interactions across the systems to understand how they impact the broader drought system across time.

**Table1: Examples of drought as a continuum in the hydrological system, ecosystem and social system based on specific studies.**

|  | hydrological system | ecosystem | social system |
|---|---|---|---|
| **antecedent conditions** | groundwater droughts more severe & spatially-coherent with dry antecedent conditions (Van Loon et al., 2017) | ecosystems affected by drought have lower NPP values under dry antecedent conditions (Machado-Silva et al., 2021) | most vulnerable to drought are the already poor & marginalized groups (King-Okumu et al., 2020) |
| **response times / resilience** | catchments with permeable geology have longer drought response time (Barker et al., 2016) | stomatal regulation gradually leads to loss of hydraulic conductance, which over time can lead to mortality (Hammond et al., 2019; Tombesi et al., 2015) | different response times of public & private sectors (Teutschbein et al., 2023) |
| **recovery** | hydrological drought recovery depends on catchment characteristics & human influences (Margariti et al., 2019; Parry et al., 2016) | ecosystem recovery times associated with ecosystem types & drought characteristics (Schwalm et al., 2017) | financial & political processes prevent social recovery (Pribyl et al., 2019) |

**3 Drought as a continuum in different systems**

**3.1 Hydrological system**

The emphasis on drought as a hydrological extreme event has led to drought detection and definition using indices specified over defined timescales (McKee et al., 1993; Mishra and Singh, 2010) or considering a limited range of lagged hydro-meteorological variables (Mishra and Singh, 2011). However, it is increasingly recognised that hydrological droughts result
from a complex interaction between multiple bio-physical processes and human influences (Van Loon et al., 2016). This implies that hydrological droughts are not singular events, but rather occur as a result of the continuous evolution of multiple hydrological fluxes and states. Therefore, we cannot fully characterise droughts without considering the (wet and dry) hydrological conditions that either precede or follow what is considered a drought event, as well as how these baseline conditions may be shifting over time due to climate change. The duration for which these hydrological conditions need to be



taken into account to understand the evolution of drought and subsequent recovery primarily depends on the processes that contribute to catchment memory (Stoelzle et al., 2020).

Catchment memory, in the context of drought, modulates the cumulative effects of anomalous meteorological and hydrological conditions and their persistence over time, and thus the severity, duration and recovery of droughts (Alvarez-Garreton et al., 2021). This memory depends on the heterogeneous and spatially distributed characteristics of the catchment,

such as topography, land cover, soil types, storage properties, and hydroclimatic conditions (Cranko Page et al., 2023; Fowler et al., 2020; De Lavenne et al., 2022). For instance, catchment memory in surface-water-dominated catchments may be quite short, depending on soil moisture and vegetation memory (Ghajarnia et al., 2020; Gu et al., 2023; Fig. 1a, dark blue line). By contrast, in groundwater-dominated catchments, catchment memory may typically be longer, as groundwater acts as a storage reservoir that buffers short-lived rainfall anomalies and sustains baseflow in rivers and streams (Sutanto and Van

Lanen, 2022). Such a long memory will, however, lead to slower recovery, particularly if groundwater levels have been significantly depleted due to more persistent rainfall deficits (Fig. 1b, light blue line). This was found during the 2018-2022 drought in groundwater-dominated systems in the eastern part of the Netherlands, which showed minimal or no recovery despite the drought being interspersed with relatively wet conditions in the winter of 2019-2020 (Brakkee et al., 2022; see 'Rhine River Basin' case study; Appendix 5). A large sub-surface storage tends to attenuate the effects of variability of

precipitation and evpotranspiration on the hydrological system, but also contributes to the accumulation of drought deficits and the lagging and pooling of meteorological drought events, thus extending the recovery process (Sutanto and Van Lanen, 2022). Other forms of storage can also contribute to long catchment memory, such as extensive wetlands and lakes (Gu et al., 2023). Furthermore, human-made storage reservoirs can increase catchment memory and buffer drought (Ribeiro Neto et al., 2022), though only up to certain critical thresholds such as when the reservoir falls empty (Rangecroft et al., 2019; Fig.

1a, orange line). Recovery in such reservoir-influenced catchments may be slower (Margariti et al., 2019; also see 'Northeast Brazil' case study; Appendix 3).

Catchment memory varies across different climate types. In arid and semi-arid climates, propagation from meteorological to hydrological drought may be slower than in wet-tropical climates (Gevaert et al., 2018; Odongo et al., 2023). This may be exacerbated by land-atmosphere interactions, which can lead to the self-propagation of droughts and thus extending them in

space or time (Miralles et al., 2019; Schumacher et al., 2022). Catchment memory also varies in climates with distinct seasonality, such as tropical savannas, snow-dominated catchments, or Mediterranean-type climates (Gevaert et al., 2018; Seager et al., 2019) where drought propagation has a strong intra-seasonal (De Lavenne et al., 2022) or even multi-annual timescale (Gevaert et al., 2018). For example, in the Andean Cordillera, snow deficits lead to streamflow deficits not only during the summer melting season but also in the following autumn season (Alvarez-Garreton et al., 2021; see 'Chile' case

study; Appendix 1). Similarly, winter snow droughts in the snow-dominated catchments of the Alps affect summer discharges of the River Rhine (Ionita and Nagavciuc, 2020; Khanal et al., 2019), while in the winter of 2022-2023, unprecedented dry and warmer-than-normal conditions over the Italian Alps caused critical hydrological conditions in the Po and Adige rivers in the ensuing spring (Colombo et al., 2023). Another example is the Mediterranean region, where



precipitation is highly seasonal due to winter storms. The weakening of the storm systems combined with long-dry summers
leads to precipitation deficit and thus, increased drought risk in the region (Cook et al., 2014; Ionita and Nagavciuc, 2021).
Catchment memory can, thus, connect climate and hydrological anomalies across different temporal scales. Precipitation
anomalies occurring at a particular time of the year can compound and lead to long-memory streamflow anomalies later in
the year (Mudelsee, 2007). Figure 1 schematically shows how the superposition of different drought signals and hydrological
states with long and short memory may result in either amplifying or dampening the duration and severity of hydrological
droughts. However, the interaction of these signals is not always linear, as witnessed by the unexpected quick recovery in
groundwater systems in Germany (Tijdeman et al., 2022: see 'Rhine River Basin' case study; Appendix 5).

It is worth noting that the processes that constitute catchment memory are not stationary. Persistent climatic deficits may
alter how a catchments responds to precipitation and/or changes in connectivity between surface and groundwater (Fuchs et
al., 2019; Fig. 1b, dashed line), leading to persistent shifts in rainfall-runoff relationships (Eltahir and Yeh, 1999; Kleine et
al., 2021) and less runoff than expected from previously similar states (Alvarez-Garreton et al., 2021; Fowler et al., 2020;
Saft et al., 2015). For instance, the Colorado River Basin in the southwestern US, is undergoing aridification as a result of
climate change (Overpeck and Udall, 2020), which then affects the characteristics of droughts through mechanisms such as
greater atmospheric water demand, increased evaporation, and lower soil moisture. Further, catchments may not always fully
recover and return to their original states after protracted droughts end, leading to new low-flow persistent states due to
changes in dominant hydrological processes and catchment memory (Peterson et al., 2021; Fig. 1b, yellow line). Alterations
in catchment memory can also arise from changes in long-term climatic patterns. For example, climate warming may
progressively shift the hydrologic regime of a basin from snow-dominated to rainfall-dominated. Increased winter
precipitation and a shift from snow to rain is dampening winter droughts, while amplifying spring and summer droughts, as
observed in Sweden (Arheimer and Lindström, 2015; Teutschbein et al., 2022) and the western US (Siirila-Woodburn et al.,
2021). Persistent climate anomalies also affect snow and glacier dynamics and storages and therefore the drought buffering
effect of these (van Tiel et al., 2023). Climate warming leads to higher evapotranspiration rates and consequently
aridification with associated changes in hydrologic regimes and drought propagation (Boisier et al., 2018; Overpeck and
Udall, 2020).   Finally, human activities can change catchment memory processes, such as through overexploitation of
groundwater, leading to depletion and degradation of aquifer systems, and causing permanent loss of natural water storage
and land subsidence due to compaction (cf. San Joaquin and Central Valley in California (Ojha et al., 2018; Smith et al.,
2017), and in agricultural basins in Chile (Fig. 1, yellow line; see 'Chile' case study; Appendix 1). Also land use change can
result in changes in catchment memory. For example, large-scale tree restoration can result in both more and less water
availability dependent on the balance between increased evapotranspiration and increased precipitation (Hoek van Dijke et
al., 2022). Similarly, the effect of urbanisation on streamflow drought is a balance between decreased water storage due to
increased imperviousness and increased water storage due to increased sewage return flows and pipe leakage (Van Loon et
al., 2022).





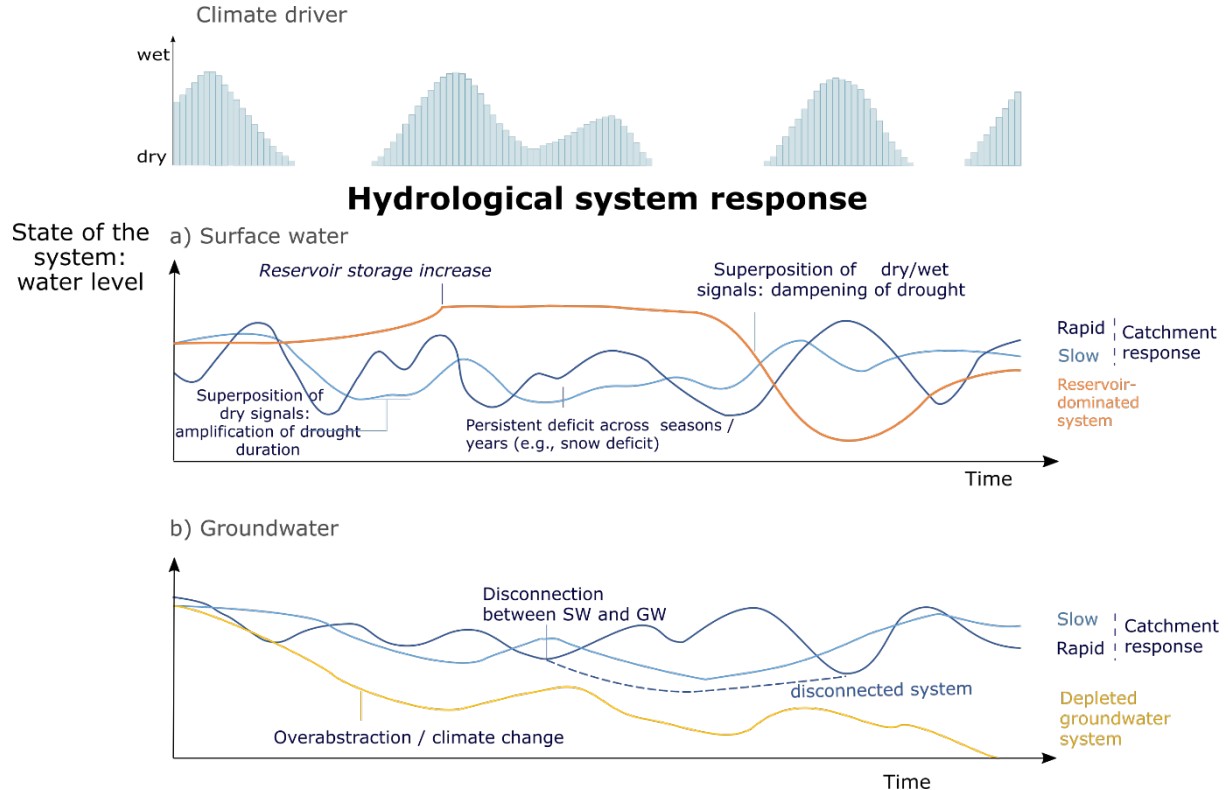

**Figure 1: Drought in surface water (SW; a) and groundwater (GW; b) in different catchments in response to a climate driver (e.g. precipitation, recharge), with fast or slow catchment response (dark and light blue lines), with a reservoir (orange line) and with groundwater depletion (yellow line). The ''drought wave'' reduces in amplitude but increases in wavelength as the catchment storage increases. This results in a superposition of anomaly signals in the hydrological system.**

### 3.2 Ecosystem

Drought has widespread impacts on terrestrial ecosystems globally and is a major driver of variability in the global carbon cycle (Ray et al., 2015; Reichstein et al., 2013; Schwalm et al., 2017; Stocker et al., 2019). The impacts of droughts on ecosystems depend on drought characteristics, ecosystem memory, and the interactions between ecosystems and their environment, and are not necessarily detrimental (Cranko Page et al., 2023; Kannenberg et al., 2020; De Long et al., 2019; de Vries et al., 2023; Wu et al., 2022). For example, plants in many semi-arid and arid systems have developed drought-tolerance traits, such as deep roots, thick and leathery leaves, drought deciduousness, and high fire resistance, allowing them to cope with drought and cascading impacts such as wildfires (Blumenthal et al., 2020; Jacobsen et al., 2008). Moreover, droughts can promote development and retention of biodiversity (e.g. in drought- and fire-dependent ecosystems) (Agee, 1996). Drought can also stabilize the ecological landscape and reinforce community resilience (Lloret et al., 2012; Fig. 2, orange line). However, shifts in drought patterns, e.g. due to anthropogenic climate change, compounded with other human pressures on ecosystems, can alter ecosystem composition and functioning, in turn affecting ecosystem resilience and potentially leading to critical legacy effects (Bastos et al., 2023; Crausbay et al., 2020; Kannenberg et al., 2020).



The impacts of drought on ecosystems take place at multiple timescales, from very short (hours to days) to very long (years to decades; Fig. 2, dark blue and light blue lines). The effects of drought on ecosystems can vary depending on vegetation type (Ruehr et al., 2019), soil type (Buttler et al., 2019), historical and concurrent climate (Ruiz-Pérez and Vico, 2020; de Vries et al., 2023; Zipper et al., 2016), microclimate (Suarez and Kitzberger, 2008), preconditions (Bastos et al., 2020) and timing of drought within growing seasons (Hahn et al., 2021; Iizumi et al., 2018) due to distinct phenological sensitivity to climatic conditions (Wu et al., 2021). The initial impacts are on plant physiological processes (Hsiao, 1973), as low water availability reduces turgor pressure in leaf cells, stomatal conductance, and xylem conductivity, and generally results in decreases in whole-plant hydraulic conductance affecting water and carbon exchanges through the soil-plant-atmosphere continuum (McDowell et al., 2022; Tyree and Ewers, 1991). These effects cascade to affect overall plant primary productivity (Griffin-Nolan et al., 2018), growth (Kannenberg et al., 2019), carbon allocation (Hartmann et al., 2020), as well as plant-plant (Zhang et al., 2019) and plant-insect (Kolb et al., 2016; Öhrn et al., 2021; Raderschall et al., 2021) interactions, or even cause plant death.

Drought can furthermore negatively affect soil carbon (C) storage by reducing belowground plant C inputs and altering their quality (Fuchslueger et al., 2016; de Vries et al., 2019; Williams and de Vries, 2020), reducing microbial activity and depleting decomposition of soil organic matter, and affecting plant and microbial communities and their interactions (Schimel, 2018). Experiments in grasslands have shown rapid responses of plant and microbial growth and community composition to drought, but slow responses of total soil C pools (Aanderud et al., 2015; Placella et al., 2012; de Vries et al., 2016). Twenty years of chronic summer drought caused persistent shifts in soil fungal and bacterial communities, and reduced microbial biomass and soil C under grasses but not under heather plants (Gliesch et al., 2024). This highlights the role of plant community composition and drought characteristics in affecting  soil C pool dynamics at different temporal scales (Fig. 2, blue line).

While some ecosystems seem to recover quickly after a single dry period, others take two or more years to recover (Anderegg et al., 2015; Schwalm et al., 2017; Wu et al., 2022). Vegetation sensitivity to drought has been reported to increase in the season following an initial dry period in some ecosystems (Bastos et al., 2021; Machado-Silva et al., 2021; Nagavciuc et al., 2023; Wu et al., 2022). These legacy effects can be caused by several mechanisms. First, plant activity recovers in several hours to years, depending on water stress characteristics (e.g., intensity and duration), the vulnerability of the plant tissues, and memory in the soil-plant-atmosphere system (e.g. from previous periods). Photosynthetic processes generally recover quickly (e.g., within hours or days) but hydraulic damage or failure require longer times, provided recovery is possible at all (Adams et al., 2017; Choat et al., 2018; Ruehr et al., 2019). Also the amount of precipitation after the drought period plays a role. For example, in the Yangtze River Basin (China), grasslands recovered already with 50% of normal precipitation, while forests required at least near-normal precipitation to fully recover (Huang et al., 2021). Second, drought reduces plant uptake of soil nutrients, leaving a larger nutrient pool available for post-drought plant growth than that in normal conditions. This could lead fast-growing plants to proliferate and decrease the ecosystem ability to cope with a second drought (de Vries et al., 2018). Third, in natural ecosystems, the cascading short- or medium-term effects of droughts





can increase background mortality (McDowell et al., 2022) and shift plant functioning and drought strategies, with long-term altered species composition and sensitivity to climate (Crausbay et al., 2020; Griffin-Nolan et al., 2019; Fig. 2, green line). Drought also shifts the composition of soil microbiota, including the balance of mutualists and pathogens, potentially leaving plants more vulnerable to subsequent drought events (de Vries et al., 2023). Fourth, chronic drought can cause reduced soil carbon inputs and a loss of soil carbon and changes in soil physical properties (Zhang et al., 2018), reducing the soil water holding capacity and rendering ecosystems more vulnerable to subsequent drought. These post-drought legacy effects

propagate to the whole plant and ecosystem, represented as changes in ecosystem functioning beyond the current growing season (e.g., defoliation detected by aerial surveys in Meddens et al., (2012), and large-scale satellite-sensed vegetation greenness by Wu et al., (2022)), losses in woody biomass for years ahead (Anderegg et al., 2015), and post-drought tree mortality or major die-off for the low-resilience ecosystems (Allen et al., 2010, 2015; Fig. 2, yellow line). A case in point are the recent droughts in Central Europe, where strong legacy effects during a multi-year drought caused massive vegetation

die-off (see 'Rhine River Basin' case study; Appendix 5). At ecosystem scale, divergent impacts and recovery responses in more diverse systems might result in weaker drought legacy effects compared to less diverse and more vulnerable systems (Yu et al., 2022). Some of these mechanisms play a role also in ecosystems dominated by annual plants, such as agroecosystems, altering vulnerability to subsequent droughts (Renwick et al., 2021), but no legacy effects of drought on crop production appeared at a national scale (Lesk et al., 2016).

In managed ecosystems, like agroecosystems and managed forests, the effects of droughts are compounded with management strategies and land use practices. For example, the vulnerability to water stress is reduced by irrigation (Luan and Vico, 2021; Zipper et al., 2016) (Fig. 2, yellow dashed line). It might also be reduced by species diversification in space (e.g., species-rich grasslands, mixed-species forests, intercropping, and drought-resistant species; (Grossiord et al., 2020; Haberstroh and Werner, 2022; McCarthy et al., 2021; Sears et al., 2021; Wright et al., 2021) or time (i.e., crop rotations;

(Bowles et al., 2020; Marini et al., 2020; Renwick et al., 2021), but exacerbated by forest clear-cut through land-atmosphere interactions (Pongratz et al., 2009; Wu et al., 2017; Fig. 2, light blue dashed line). Soil management, e.g. tillage, affects soil properties and functioning and hence the response to drought, in ways depending on local conditions (Pittelkow et al., 2015; Schneider et al., 2017). Time scales for implementation of mitigating actions (e.g., irrigation) or for the effects to emerge (e.g., crop rotations) can be long (Marini et al., 2020; Renwick et al., 2021). Most of the cascading effects that appear in

natural ecosystems are similar in managed ecosystems but these can be buffered or amplified by specific management practices (e.g. rotation periods, age structure, stand density, diversification). Nevertheless, in managed ecosystems, plant species composition is defined by management itself, not by plant community evolution, although still in the context of existing climate and risk avoidance preferences. Over time, increasing droughts could for example promote the adoption of climate-resilient crops and varieties (Acevedo et al., 2020).

The prolonged effects of drought could also increase the frequency of severe wildfires (see 'Chile' and 'Colorado River Basin' case studies; Appendix 1 & 2), but actual impacts on ecosystems may be delayed by years or decades depending on fire management strategy. While wildfire consequences can manifest in individual dry years (Abatzoglou and Williams,



2016; Holden et al., 2018; Littell et al., 2016), tree damage and mortality driven by the effects of earlier extended droughts increase fire severity, frequency and burned area in historically fire-adapted forests (Stephens et al., 2018). However, wet

periods are also important for wildfire risk. For example, some of the current wildfire crises in the Western US stem, in part, from a relatively wet period in the 1950's and 1960's facilitating fire suppression in historically frequent-fire forests, followed by deepening droughts and extended dry spells (Abatzoglou and Williams, 2016; Holden et al., 2018) in forests with now more abundant fuels (Hessburg and Agee, 2003). In addition to fire-induced tree mortality, tree mortality can also occur following drought because of increased risk of insect outbreaks due to the lower resistance of drought-stressed trees

(Fettig et al., 2019; Kolb et al., 2016; Luce et al., 2016; McDowell et al., 2022; Fig. 2, yellow line). As a consequence of these long-term and cascading impacts some ecosystems are now seeing extensive ecological transformation (Crausbay et al., 2020; Steel et al., 2023).

Some of these processes feed-back to the development of the hydrological drought. For example, with increasing water limitation, some plants are able to increase their water-use efficiency and therefore buffering water loss through transpiration

(Flach et al., 2018; Peters et al., 2018) and maintaining photosynthesis, although contrasting patterns have been found globally (Yang et al., 2016). Differences in water use and drought stress responses underlying different vegetation types can, therefore, contribute to asymmetries in the development of soil-moisture anomalies during drought (Bastos et al., 2020; Flach et al., 2018). Other factors such as earlier onset of the phenological cycle may further contribute to exacerbate summer drought (Lian et al., 2020).

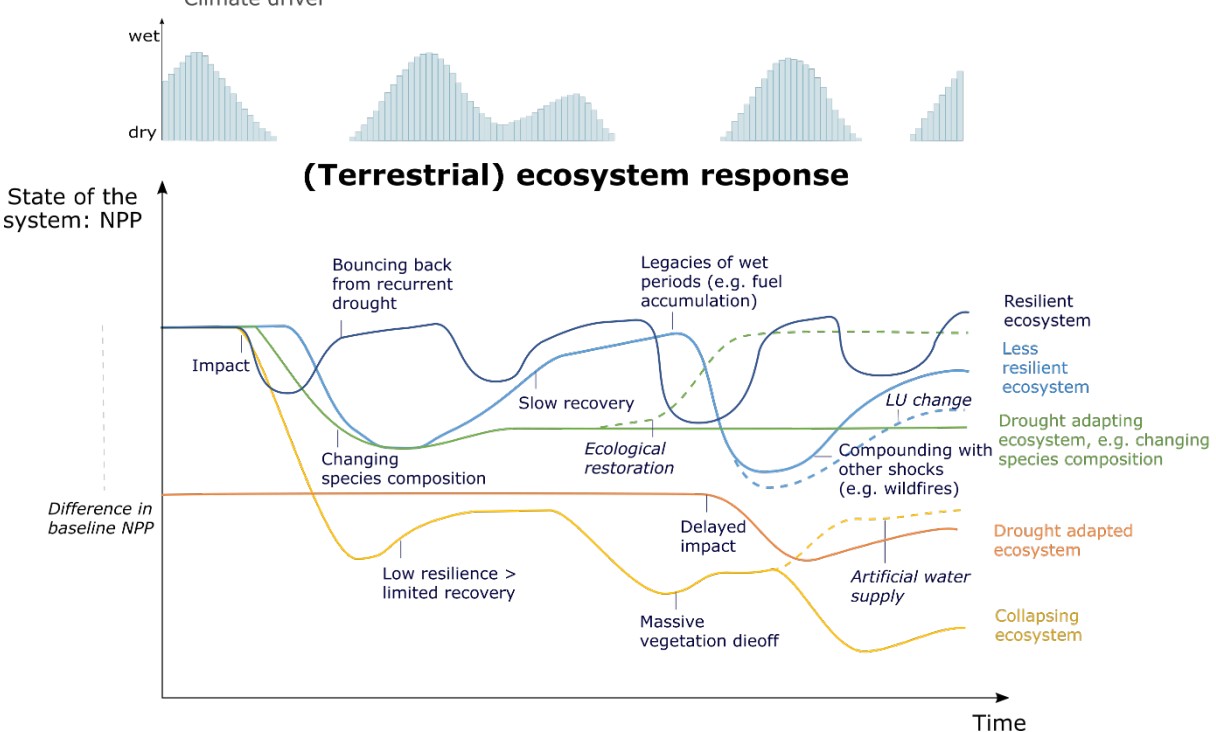




**Figure 2: Drought in a variety of different ecosystems in response to a climate driver (e.g. precipitation, recharge), with drought-adapted ecosystems (orange line) starting at a lower baseline, drought-adapting ecosystems (green line) changing in response to drought, more or less resilient ecosystems (dark and light blue lines) impacted by and recovering from drought, and collapsing ecosystems (yellow line) gradually decreasing due to recurrent droughts. NPP = Net Primary Production, LU = Land Use. The dashed lines represent ecosystem management and other human influences.**

### 3.3 Social system

Less water than normal in part of the hydrological system can have a significant negative impact on social systems, including livelihoods, food security, and health, but is only impactful when it exceeds the capacity to manage the water deficit, for example when it affects an already vulnerable population (Raju et al., 2022). Thus, social impacts of drought arise when policies, regulations and other drought management actions fail or are inadequate and society cannot cope with dry conditions. Additionally, different groups or sectors of society can be impacted by the same drought but in different ways and at different times (Stahl et al., 2016; Wlostowski et al., 2022); they may also respond differently (Teutschbein et al., 2023). For instance, rainfed agriculture is sensitive to meteorological and soil moisture drought, while hydropower and inland navigation are sensitive to hydrological droughts (Van Loon, 2015; Teutschbein et al., 2023). Consequently, the timing of drought impacts in each context is related to the speed of drought propagation through the hydrological cycle (Van Lanen et al., 2016) (Fig. 1) and societal processes (such as water allocation laws, priority rules) that can cause amplified, attenuated or lagged effects (Fig. 3a, dark blue and light blue lines). This can be observed when immediate drought impacts are delayed with prevention and mitigation measures (e.g. irrigation) but will be felt later, and potentially more severely, when the drought propagates to surface water or groundwater (van Dijk et al., 2013). Drought impacts on society can also be reduced with emergency measures like food aid (Fig. 3a, dashed line), which may not be sustainable in the long term if drought persists. In some sectors, societal responses can even increase the impacts of drought, such as when drought reduces water availability but also increases water demand, the combination of which stresses public water supplies (Di Baldassarre et al., 2018).

Similar to their propagation through the hydrologic cycle, drought impacts cascade through society and the economy with different speeds, affecting different groups and regions at different intensities and timings, and potentially far from where the drought originated (de Brito, 2021). Drought impacts are often gradual changes in factors that can also be influenced by other processes (e.g. decline in crop yield, energy production losses, reductions in goods transported), which makes it difficult to define whether anomalies are impacts of a drought or caused by something else and when they start and end (Hall and Leng, 2019). Moreover, some of these impacts can be the result of the drought itself (lower water availability) or be related to responses to drought (lower water allocation). In response to drought, individuals can take a variety of measures to mitigate impacts such as leaving some portion of agricultural land fallow, minimizing transport loads, decreasing outdoor or inessential water use, among others. Decision-makers may decide to implement water use restrictions (van Oel et al., 2018; Ribeiro Neto et al., 2022) or restrictions on navigation or cooling water discharge in order to preserve limited water supplies for more critical uses. Migration can also be seen either as a coping mechanism or adaptive measure against drought (Falco et al., 2019; Vinke et al., 2020). Migration can also increase the vulnerability of the migrating group (e.g. decreased health /




financial resources) or put extra stress on the water resources of the receiving area, potentially affecting also the original communities there. These drought-related decisions and restrictions also impact society, as they shift exposure to water deficits from one group or system at risk to another.

Drought impacts may also linger long after the drought hazard has ended (WMO, 2021), creating indirect impacts such as
disrupted international trade (Carse, 2017), temporary or permanent unemployment, business interruption (Ding et al., 2011), loss of income (Zaveri et al., 2023), mental health issues (Vins et al., 2015), disease due to poor water and air quality (Charnley et al., 2021; Mora et al., 2022), and food insecurity, malnutrition, starvation and widespread famine (Bailey, 2013; UNDDR, 2021), among others. However, drought impacts may also be positive for some groups; for example, increased crop prices may result in higher incomes for those farmers who do not suffer a significant production loss (Ding et al., 2011)
(Fig. 3a, green line), and dry and warm weather may boost tourism, especially in cold, wet climates, e.g. mountain areas (Koutroulis et al., 2018; Liu, 2016; Wlostowski et al., 2022).

How society is impacted by and responds to drought is not only dependent on a single drought, however. It is also shaped by a complex interplay of dry and wet cycles, socio-political pre-conditions, socioeconomic dynamics, adaptive behavior at multiple governance levels, and other processes directly or indirectly related to drought. In this section, we discuss three
points that illustrate how drought functions as a continuum in the social system.

Firstly, social vulnerability to drought is dynamic, both within and between droughts (de Ruiter and van Loon, 2022). Communities may be more or less vulnerable to an acute drought situation due to their existing levels of exposure, sensitivity, and adaptive capacity, and due to underlying inequalities (with regard to water access, but also who benefits from the top-down reactive drought management) (IPCC, 2014; UNDDR, 2021). Critically, however, these aspects of
vulnerability are influenced by evolving drought conditions and past droughts, among other, more external, factors. For instance, extended dry situations may gradually erode communities' financial resources (Enqvist et al., 2022; Kchouk et al., 2023; Savelli et al., 2021) (see 'Northeast Brazil' case study; Appendix 3), physical (Belesova et al., 2019; Sena et al., 2017; Treibich et al., 2022) and mental (OBrien et al., 2014) health, access to education (Hyland and Russ, 2019), family and community harmony (Dean and Stain, 2007), and more (Keshavarz et al., 2013) in ways that exacerbate ongoing or future
vulnerability (Fig. 3a, yellow line). For instance, repeated drought can deplete a household's resources, making migration or other coping/mitigation choices impossible and trapping societies in a vicious cycle of increasing vulnerability (Black et al., 2011; Black and Collyer, 2014; Nawrotzki and DeWaard, 2018).

How a community recovers after a drought event also influences future vulnerability. Societies that recover quickly after a drought have been found to be less vulnerable to the next, compared to societies that recover slowly (Di Baldassarre et al.,
2018; Kchouk et al., 2023; Weiss and Bradley, 2001). However, returning quickly to a past state without considering the need to build resilience to future events can exacerbate vulnerability and undermine long-term resilience (Koebele et al., 2020). Indeed, successive droughts, or droughts compounded by other hazards, extend the recovery time of affected communities. A static level of vulnerability therefore cannot be defined for a specific event, and using pre-drought estimates of vulnerability, and averages calculated over extended periods can underestimate compounding vulnerability.



There also is a strong imprint of long-term social, political, economic processes unrelated to drought on social vulnerability and therefore on drought risk. For example, in the 2018 Cape Town drought (Day Zero), Apartheid-era social processes influenced vulnerability to drought through historical spatial and economic segregation, which led to long-term unequal access to water (Enqvist and Ziervogel, 2019; Savelli et al., 2021) and made some communities inherently more vulnerable (Fig. 3a, different baselines). An aggressive water metering campaign by the government, coupled with massive increases in

the price of water, further strained these communities' already limited financial resources (Enqvist et al., 2022).

Secondly, adaptation happens in response to past, ongoing, and/or expected drought experiences, which influences future drought risk (Kreibich et al., 2022). While short-term coping measures, such as buying food or water, are stopped when they are not needed anymore, long-term adaptation measures, like implementing irrigation or changing livelihood, have a long legacy. Adaptation happens on the scale of individuals and communities, as well as governments, and is strongly related to

individual and collective memory. While drought events may leave a significant impact on people's memory due to the immediate and tangible effects experienced (Duinen et al., 2015; Gebrehiwot and van der Veen, 2021; Griffiths and Tooth, 2021; Taylor et al., 1988), this memory likely fades over time, especially if something else eventful such as flooding happens (Garcia et al., 2022). Recency bias in human memory (related to the availability heuristic, Garcia et al., 2022; Tversky and Kahneman, 1973) gives greater importance to the most recent events. This can lead to a gradual decrease in the perceived

risk of droughts and the neglect of long-term drought-management practices (Fig. 3a, orange line).

Communities can retain the memory of previous droughts through institutional arrangements (Howden et al., 2014), cultural practices and collective experiences (Pandey and Bhandari, 2009; Salite and Poskitt, 2019; Shiferaw et al., 2014). In drought-prone areas of Sub-Saharan Africa, for example, farmers have adopted different drought-risk coping strategies to reduce their risk of drought (see 'Kenya' case study; Appendix 4). These include choosing specific crop varieties, temporal

adjustments of the cropping calendar, change of weeding and fertilization practices, and use of soil and water conservation practices. Some of these strategies, which originated as coping mechanisms, have become an integral part of the farming system that reduces overall risk (Pandey and Bhandari, 2009; Shiferaw et al., 2014; Fig. 3a, green line).

Droughts do not only trigger individual and community action, but also management response from governments across levels. Droughts can drive short- and long-term policies and decision-making. Short-term crisis management is most

common, including emergency relief, such as water trucks and cash transfers, targeted at specific areas and affected groups (Barendrecht et al., 2024; Wilhite, 2017). However, governments often deal with each drought as a "new" or unique event, possibly because of a low memory of "creeping disasters" like drought (Ulibarri and Scott, 2019; Wilhite and Glantz, 1985) (illustrated in the hydro-illogical cycle; Fig. 3b, blue line). This makes it less likely that long-term proactive measures are implemented for drought, compared to other natural hazards. For example, in the Netherlands, severe drought events (1976,

2003) resulted in less structural measures than flood events (1953, 1993) (Bartholomeus et al., 2023). Furthermore, applying emergency measures to address the persistent impacts of droughts may conceal the need of long-term management strategies and lead to unintended consequences in other systems, such as the case of Chile, where the law has suspended the obligation to maintain ecological flows in several basins for over eight consecutive years as a means to mitigate socio-economic



drought impacts (see 'Chile' case study; Appendix 1). Long-term management starts to emerge after multiple drought events
as a lagged-effect scenario. According to Nohrstedt, (2022) this means that "transformation does not materialize as an
immediate response to dramatic agenda-setting disaster, but will rather emerge gradually through time due to accumulated
experience from multiple events" (Nohrstedt, 2022; p.432; Fig. 3b, green line). A review by (Mendoza et al., 2024) found
that in several studies, communities were able to distill insights from previous drought experiences through farmer field
schools and other collaborative learning spaces. Engaging in long-term collaborative learning enabled communities to see
which adaptive strategies worked best in specific conditions. However, management or adaptation is not only implemented
in response to drought, but can also be implemented in anticipation of (increasing) drought risk. Recently, climate change
projections, with special attention to the risks of droughts, are being actively integrated into policy development and
decision-making processes in Europe. This is evidenced by the initiatives and strategies outlined in the first European
Climate Risk Assessment (EUCRA, 2024) and the European Drought Risk Atlas (Rossi et al., 2023) through advanced
modeling, systematic risk assessments, and addressing interconnected and cross-border impacts.

Thirdly, responses to drought can later turn out to be maladaptive. Maladaptation (or rebounding vulnerability) is when the
outcome of adaptation measures ends up increasing the vulnerability of a community over time (Juhola et al., 2016;
Schipper, 2020) (Fig. 3b, yellow line). For example, increasing water storage and supply with reservoirs provides a buffer
during dry periods, but can also lead to a form of safe development paradox called the "reservoir effect" (Di Baldassarre et
al., 2018). Over-reliance on reservoirs can increase social exposure and vulnerability when a drought occurs. Short-term
adaptation measures can also erode the conditions for sustainable development by consuming the adaptive capacity of a
community and preventing it from taking measures with long-term benefits. For example, if during a previous drought, there
has been an increase in groundwater pumping that has continued after the drought, the impacts of a second drought may be
experienced more quickly due to the added effect of groundwater pumping (Pauloo et al., 2020). Moreover, changes in
rainfall patterns can affect water user behaviour, which again may influence the sustainability of small-scale rural water
service providers due to high intra-seasonal revenue variability (Armstrong et al., 2022).

Maladaptation impacts vary across society as a result of social processes including poverty, inequality, power asymmetries,
and ineffective decision-making. For example, in Cape Town during the 2018 drought, the wealthiest populations, who
already had the highest consumption rate prior to the crisis (Enqvist and Ziervogel, 2019), could also afford to implement
coping strategies such as drilling private groundwater wells (Simpson et al., 2019), which ultimately decreased their
vulnerability to drought compared to pre-crisis level but lowered water availability for those who could not afford to drill
deeper wells. In rural areas of Chile, people rely on self-organized communities with inadequate infrastructure for providing
subsistence drinking water, leading to water cuts that have been remedied by cistern trucks. Cistern trucks have become a
non-structural reactive measure to address permanent water access requirements in rural areas. On the other hand, people in
urban areas rely on water sanitation companies and have not been affected by water cuts since these companies have
adequate infrastructure (see 'Chile' case study; Appendix 1).





a)

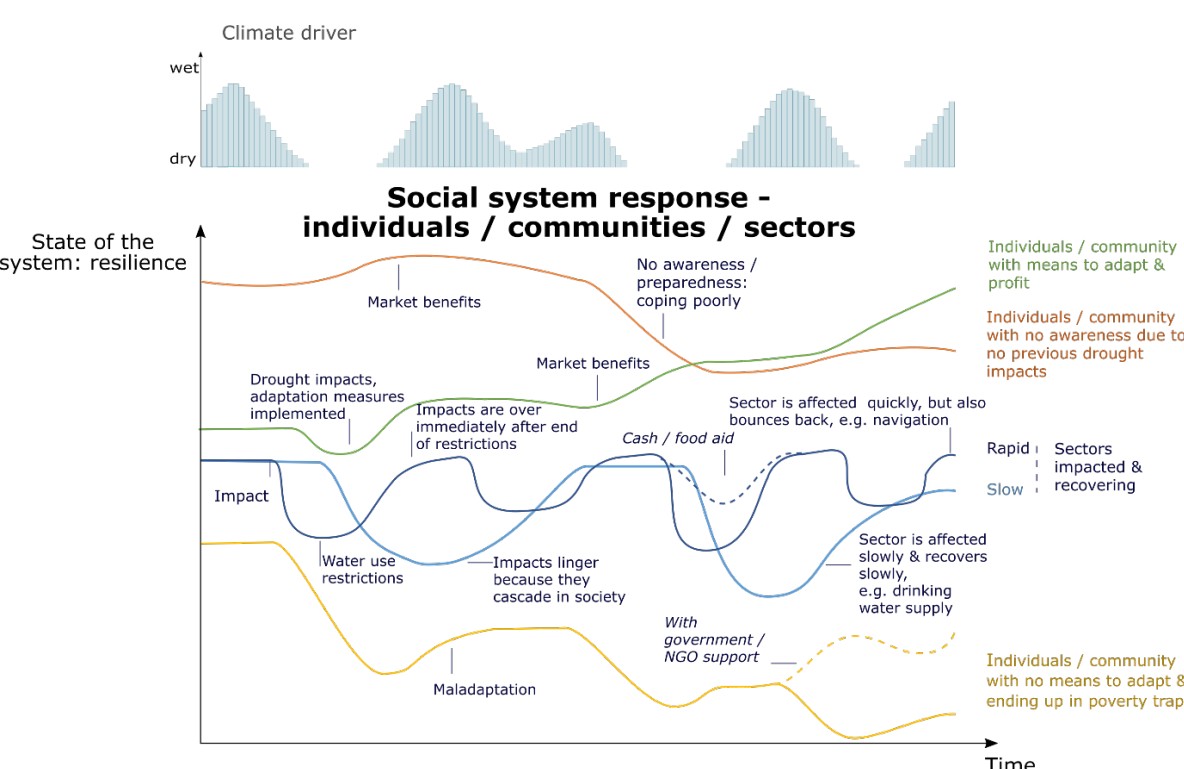

b)

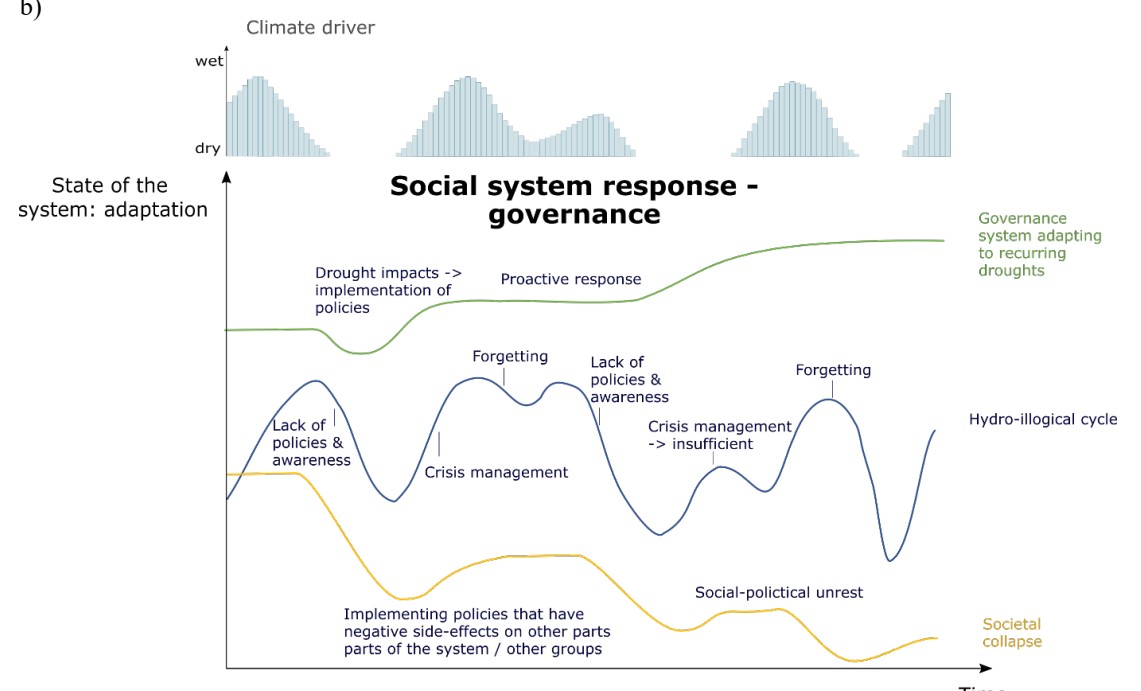





**Figure 3: Drought in the social system in response to a climate driver (e.g. precipitation, recharge): a) individuals, communities, sectors, b) governance. a) Different communities have different baseline resilience (starting point on y-axis) and different means to adapt and respond (green and yellow lines), different sectors can be impacted and recovering on different time scales (dark and light blue lines), and communities can be highly resilient, but also unaware and therefore more affected when drought hits (orange line). b) Governance systems can follow the hydro-illogical cycle (Wilhite, 2011) of emergency response and forgetting (blue line), implement policies that allow for proactive response and preparedness (green line), or create a maladaptive system that could end up in societal collapse (yellow line).**

## 4 Drought as a continuum in the system of systems

### 4.1 Similar emergent temporal patterns between the systems

In the hydrological, ecological, and social systems studied in Section 3, common patterns are visible, corresponding to the systems theory element of "self-organization and emergence" (Section 2). We see that the dynamics of drought in all systems can be characterised as a combination of different fluctuations, cycles, gradual changes, and shocks that emerge from memories and responses within the specific system, following the systems theory element of "resilience and adaptation" (Section 2). This leads us to define a typology of the drought continuum, with four archetypical temporal drought trajectories (Fig. 4):

1. **Impact & recovery**: the system is affected by drought but subsequently bounces back. Depending on the type of system this impact and recovery can happen quickly or slowly, related to short or long memory (type 1a and 1b, Fig. 4). Superposition of signals with different timeframes can occur, like we discussed for the hydrological system (Section 3.1).

2. **Slow resilience building**: the system adapts well to drought and drought resilience increases over time. We see this for example in selected social systems (e.g. drought-resilient farming systems; Section 3.3) and in ecosystems (e.g. drought-tolerance traits; Section 3.2).

3. **Gradual collapse**: the system becomes more vulnerable with each drought and changes to a negative state. In the ecosystem, we see this as a result of long-term legacy effects and compounding processes (Section 3.2). In the social system, this happens as a combination of a high baseline vulnerability and maladaptation (Section 3.3).

4. **High resilience, big shock**: the system has an initial highly-resilient baseline and is not affected initially, but is impacted more after a prolonged or successive droughts because of lack of awareness and preparedness. We see this most clearly in the social system, when low vulnerability leads to low awareness until a severe drought causes a tipping point (Section 3.3).



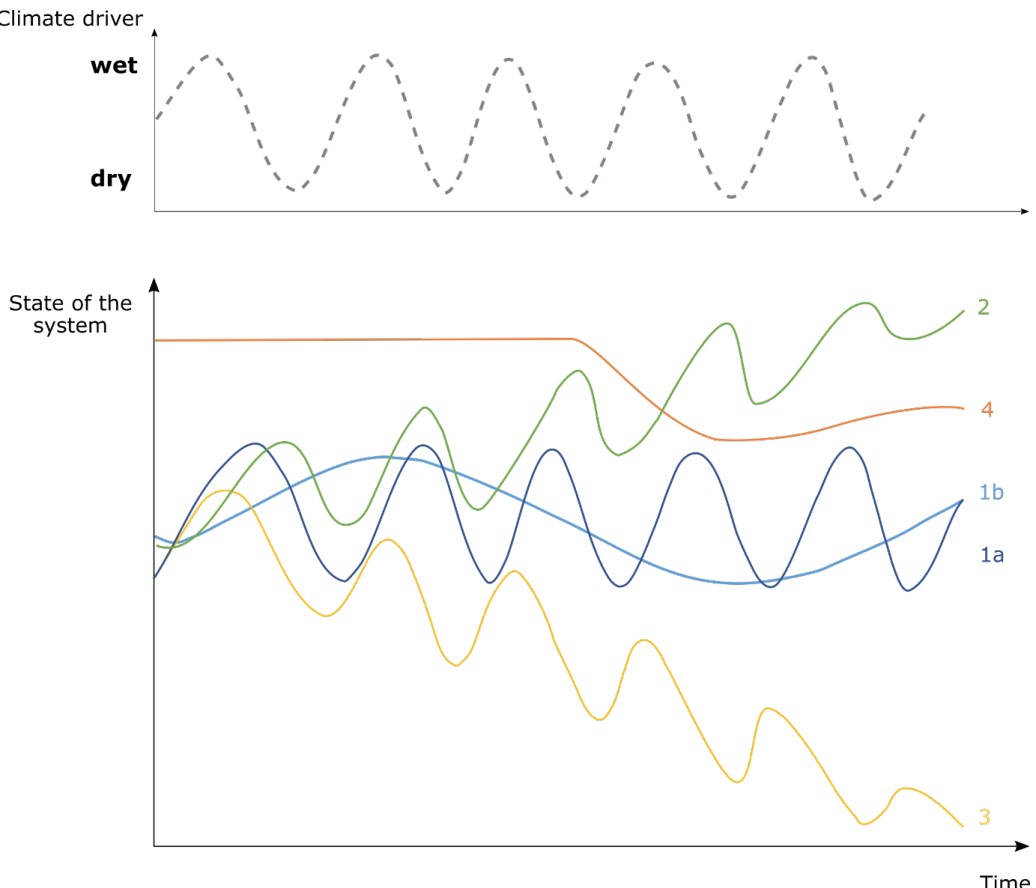

**Figure 4: Typology of drought continuums emerging from the analysis of temporal drought trajectories in the hydrological, ecological and social systems; (1) impact & recovery, (2) slow resilience building, (3) gradual collapse, and (4) high resilience, big shock.**

## 4.2 Systems interactions leading to critical transitions overflowing between systems

Because the systems are intrinsically intertwined, a change in one system leads to a change in another system. This can trigger a system to change trajectory, leading to type-transitions. These interactions correspond to the systems theory elements of "non-linear behaviour and tipping points" and "state shifts and feedback loops" (Section 2).

For example, a **High resilience, big shock** social system may be using water resources unsustainably and depleting groundwater, shifting the hydrological system from an **Impact & recovery** system to a **Gradual collapse** system (point 1, Fig. 5a). At first, this may bring benefits to society and not impact the ecosystem too much, but at a certain moment a tipping point is reached where the ecosystem also moves into **Gradual collapse**, for example when groundwater dependent ecosystems dry out completely and are lost (point 2, Fig. 5a).

On the other hand, an **Impact & recovery** social system that implements reservoirs can create a **High resilience, big shock** hydrological system, but at the same time can cause a **Gradual collapse** in the ecosystem (point 1, Fig. 5b). If society then



shifts to a **Slow resilience building** social system, such as through adoption of nature-based solutions, that may nudge the hydrological system into **Impact & recovery**, then the ecosystem can also evolve to **Slow resilience building** (point 2, Fig. 5b).


a)

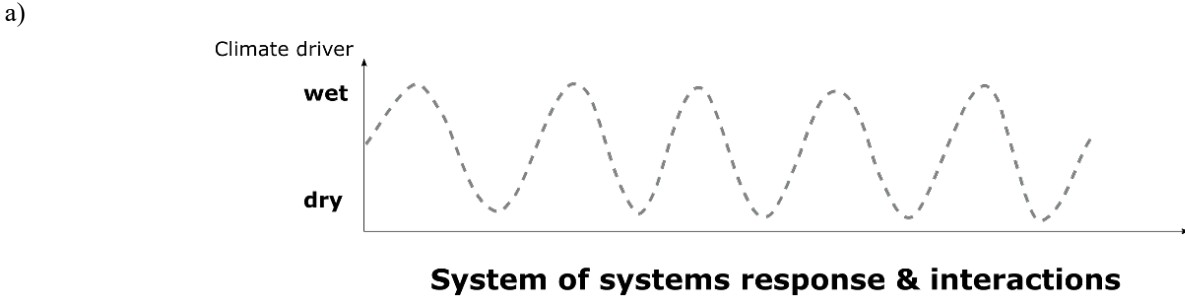

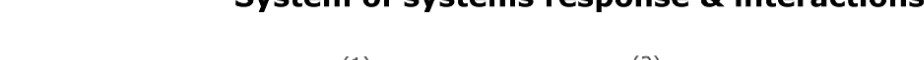

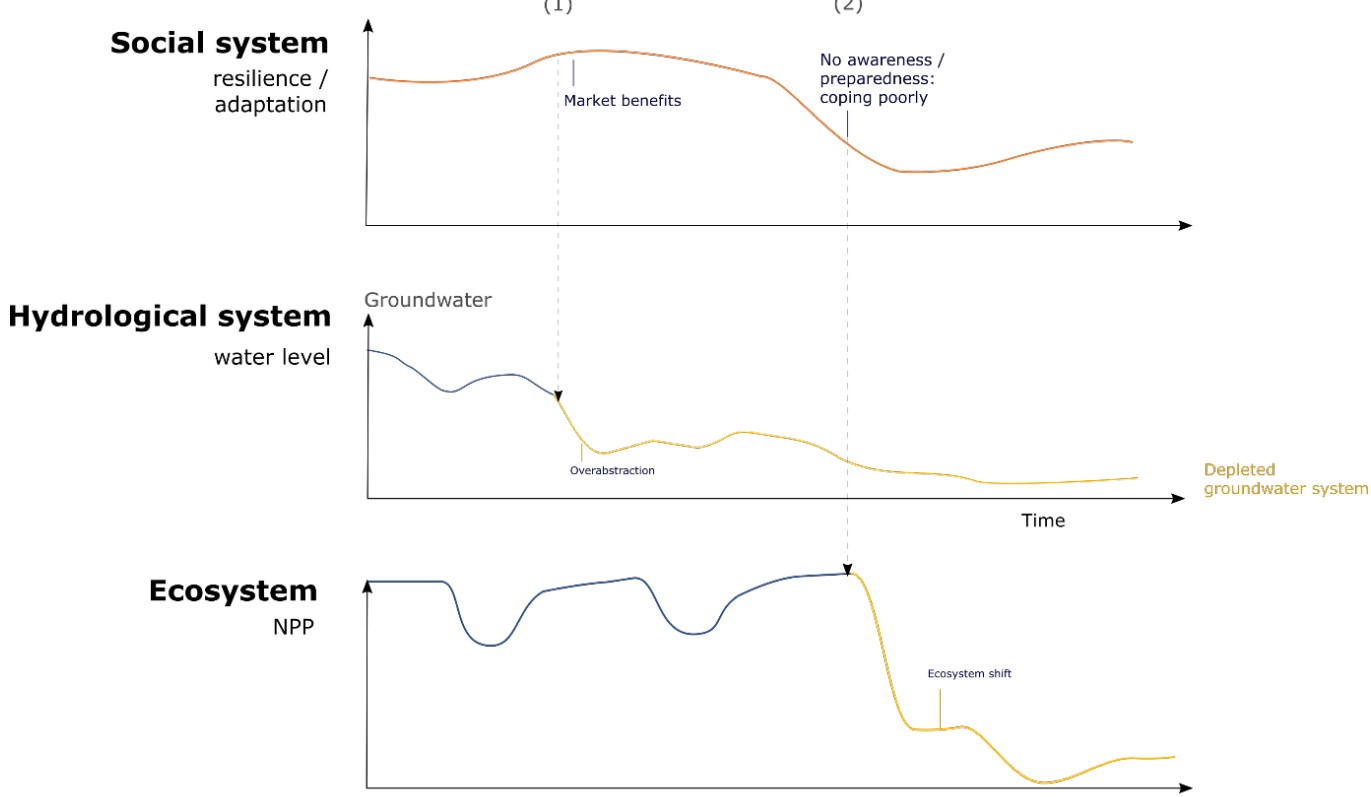




b)



**Figure 5: Example pathways of connected systems moving between types: (a) From an impact & recovery system to a gradual collapse system, and (b) from an impact & recovery system to a slow resilience building system.**

**4.3 Case studies of hydro-eco-social drought continuums**

We explored different drought typologies, system-interactions and type-transitions in five case studies (Chile, Colorado River Basin, Northeast Brazil, Kenya, Rhine River Basin; see Appendix 1-5).



### 4.3.1 System types

System types are apparent in the case studies. For example, in Kenya (Appendix 4), we see the **Impact & recovery**
typology. Short heavy rainfall has been demonstrated to play an important role in groundwater recharging after drought,
resulting in recovery of the hydrological system (Matanó et al., 2023). The case in Northeast Brazil (Appendix 3) shows
signs of a **Gradual collapse** typology due to maladaptation. In some communities, the introduction of a reservoir aimed at
reducing drought vulnerability, ultimately proved to be maladaptive. As the reliance on external labour for income shifted to
intensive irrigated agriculture following the reservoir's implementation, the community's financial resources progressively
eroded due to recurring droughts (Kchouk et al., 2023). And in the Rhine River Basin (Appendix 5), we find an example of
the **High Resilience, big shock** type. The system is highly managed and engineered, vulnerability is low and there is a lack
of awareness, which means that drought impacts are often not felt by society. Impacts are compensated due to market price
effects or are passed on in time or passed down to other systems. Drought management ignored connections between
systems and longer-term impacts, until this was no longer possible in the recent droughts in 2018-20 (Bartholomeus et al.,
580    2023).

### 4.3.2 Systems influencing each other

We also see examples of one drought system influencing another. The "megadrought" in Chile (Appendix 1) instigated
changes in the hydrological system, which were influenced by the social system. During the mega-drought snow-dominated
basins in the Andes generated on average 30% less streamflow than previous single-year droughts, due to long catchment
memories (**Impact & recovery – type 3b**). In some rural areas, this memory effect leading to lower flows has overlapped
with water extractions for human activities in downstream sections (**Impact & recovery – type 3a**), leading to an
amplification of drought signals in lowlands, resulting in the drying out of lakes and pumping wells supplying water for
human consumption (**Impact & recovery – superposition of signals due to interaction hydro & socio**). Also the drought
response of ecosystems was strongly influenced by the social system. Before the megadrought, wildfires were concentrated
between November and April, but now they extend from October to May, increasing the occurrence period from 6 to 8
months (**Impact & recovery – change of type 3a to 3b due to interaction hydro & eco**). More than 70% of the megafires
(>50,000 ha of burned area) of the last four decades have occurred during the megadrought, where 50% of the burned area
corresponds to industrial tree plantations. It is also worth noting that 99% of wildfires in Chile are caused by human actions,
whether they are accidental or intentional. In these examples, the system maintains its typology and the interactions cause a
lengthening or enhancing of the drought memory leading to changes in response and recovery.
This was also apparent in the Colorado River Basin (Appendix 2), which has been in a "mega-drought" for over two decades
(2000 to present). Despite punctuated periods of high winter precipitation, the basin has experienced more very dry years
than normal and major reservoirs have been drawn down to record low levels. Here, the hydrological system strongly
influences the ecosystem. Extended drought, coupled with aridification, threatens the health of various riparian environments





and endemic species, some of which already face a risk of extinction (**Gradual collapse – speeding up due to interaction hydro & eco**). It may also lead non-native species to dominate over native species (Rogosch et al., 2019). The hydrological system has also influenced the social system, especially governance. In the early years of the drought, policy makers took incremental actions because they did not proactively estimate or understand how severe the drought would get. This has led to the need for repeated policy changes, which challenges policy maker and stakeholder capacity, as well as public support

(Deslatte et al., 2023; Garcia et al., 2020) (**Impact & recovery – type 3a**). The occasional "wet" year also may have undermined drought adaptation as decision-makers "forget" or "ignore" the drought (**Impact & recovery – less memory due to interaction hydro & socio**), though these years may also provide helpful buffers to make adaptation measures successful (e.g. adding a small amount of water to storage reservoirs). The highly polycentric governance system of the basin – meaning that many different actors and sectors have authority to make decisions about water management – may also

challenge drought governance due to actors' different values, legal rights, and experiences of drought.

### 4.3.3 Feedback between systems

Feedback between systems results in a two-way influence on memory. The high concentration of small dams in Northeast Brazil (that have been built to protect against drought; Appendix 3) affect the memory of the watershed. During multi-annual droughts, these dams remain dry longer due to low precipitation and high evaporation rates. This in turn reduces

hydrological connectivity by decreasing runoff and recharge to the large reservoirs. These reservoirs are vital for urban water supply, and the delay in recharge prolongs the impacts of the drought (Ribeiro Neto et al., 2022). So this is an example of how the hydrological system changes from **Impact & recovery type 3a to 3b** due to the interaction between social and hydrological systems, and how then the social system also changes from **Impact & recovery type 3a to 3b** due to the interaction between hydrological and social systems.

Feedback between two systems can also result in (potential) type-transitions, for example the social system causing a **Gradual collapse** of the hydrological system, which then triggers a **High resilience, big shock** in the social system. In Chile (Appendix 1), a large portion of the drinking water supply in the capital is obtained from long-memory groundwater systems that are consistently being depleted and that may already have been disconnected from subsurface flows and recharge (e.g., 300-m pumping wells were recently inaugurated as a key drought adaptation strategy). This generates a false perception of

not being under drought since, despite a decade-long megadrought and depleted surface reservoirs, people are not experiencing water shortages in Santiago. This may lead to increased vulnerability - the opposite aim of the adaptation strategy - by over-relying on a water supply system that is based on an invisible reservoir that has an unknown but finite volume which is not being replenished. A similar situation emerged in the Colorado River Basin (Appendix 2), where, due to high levels of natural interannual variability, major reservoirs were constructed in the previous century to store water from

snowmelt for use in dry years (i.e. "buy time" during drought years). However, this response may be maladaptive in the future as the climate changes (more dry years, higher temperatures, more precipitation as rain rather than snow), leading to other unsustainable actions that forebode **Gradual collapse** (i.e. increase groundwater pumping when surface water is not



available) (Garcia et al., 2020). This has already led to significant water supply challenges, especially in the lower half of the basin. However, sustained collaboration among policy actors on water sustainability may push the social system toward a
mode of **Slow resilience building** (Karambelkar and Gerlak, 2020; Koebele et al., 2020).

### 4.3.4 Dynamic shift of the hydro-eco-social system

Interactions between all three subsystems can result in a dynamic shift of the hydro-eco-social system. In Kenya (Appendix 4), the more frequent droughts in the 21st century pose new challenges for the social, ecological and hydrological systems. Key social processes affecting land use, most prominently agricultural expansion, have affected hydrological and ecological
systems. Land use changes have affected how meteorological drought propagates to hydrological drought, and have led to a weakening of ecological buffers to drought, such as riparian forests (**Impact & recovery – change of type 3b to 3a due to interaction hydro & eco**). While policy responses to drought in Kenya have historically been reactive, there has been an emergence and expansion of public dams, water reservoirs and irrigation systems for crop-farming in previously pastoral areas. While being able to buffer for droughts (**Impact & recovery – change of type 3a to 3b due to interaction hydro &**
**socio**), new socio-hydrological dynamics are triggered, such as reservoir effects (**High resilience, big shock**), divergent paths of vulnerability among water infrastructure users, and pressure on surrounding natural resources (**Gradual collapse**). Continuously adapting to the new drought reality, a young generation of pastoralists are starting to support more strict grazing zone management, which may reduce degradation of vital hydrological and ecological buffers to drought (**Slow resilience building**). This is an example of where shifts in the social system can trigger type-transitions so that the combined
hydro-eco-social drought system can move to a **Slow resilience building** typology.

Similarly in the Rhine River Basin (Appendix 5), historically, (ground)water levels and vegetation interact dynamically and have been able to recover from shocks (**Impact & recovery**). Over time, artificial drainage and over-abstraction have resulted in depleted groundwater, and agricultural management and pollution has rendered ecosystems highly vulnerable. The combination of these factors has led to a stressed system that was impacted strongly in the 2018-20 drought and showed
very limited recovery (**Gradual collapse due to effect of socio on hydro & eco**). However, this recent event also sparked interest and awareness, which resulted in improvements in drought monitoring and forecasting, in the development of new policies, and implementation of more sustainable adaptation (nature-based solutions). For example, in the Netherlands, regional water authorities can implement surface water use restrictions during drought, but after the multi-year drought, there is an increased awareness that groundwater use should also be restricted with the aim to prevent long-term effects in the
hydrological system and potential cascading effects on the social system and ecosystem (Bartholomeus et al., 2023). These are the first signs that the **Gradual collapse** is being changed to **Slow resilience building**, in which changes in the social system are improving the hydrological system and the ecosystem, finally benefiting the entire hydro-eco-social system.



## 5 Outlook

The use of systems theory to explore the temporal dimensions of the hydro-eco-social drought continuum has provided important insights. These insights could be used in future studies and practices to improve drought management. Here, we discuss some suggestions.

### 5.1 Scientific outlook

Research on drought as a continuum should encompass both enhanced process understanding and improved tools and methods. We suggest that:

1) Hydrological modelling tools used for drought analyses should better represent memories in the hydrological system. These memories were found to contribute to a better forecast skill for streamflow drought (Du et al., 2023; Sutanto et al., 2020; Sutanto and Van Lanen, 2022), but not all hydrologic memories are currently represented well in modelling tools, especially multiannual dynamics (Fowler et al., 2020) and catchment memory processes related to snow, groundwater (Tallaksen and Stahl, 2014) and vegetation (Troch et al., 2013). Further, analyses and

predictions should incorporate non-stationarity in hydro-eco-social processes and future extremes (Brunner et al., 2021; Samuel et al., 2023). Substantial improvement can come from better incorporating these dynamics.

2) Modelling of ecosystem dynamics and memory should also be further developed. State-of-the-art process-based ecosystem models already mostly include soil water dynamics and some of its delayed effects on physiological processes, but should also consider longer-term key legacy effects of droughts and other disturbances, via for

example explicit consideration of groundwater dynamics (Mu et al., 2021), drought-induced structural damage (defoliation, xylem damage) and mortality and hence ecosystem composition or enhanced vulnerability to pest (Kolb et al., 2016) and wildfire (Hantson et al., 2016; Luce et al., 2016). Similarly, soil-mediated long-term legacy effects, e.g. via microbial community composition and activity, and soil carbon effects on soil hydraulics or interactions with nutrient availability, are generally neglected. Generally, these modeling limitations arise from

currently limited mechanistic understanding of these processes, especially at regional to global scales.

3) Analyses of the social processes underlying drought risk should better include temporal dynamics and effects of social memory for individuals, communities, and governance systems. Given the role of socioeconomics, inequalities, perceptions, and other social processes in defining drought risk and recovery, the ability of governance systems to maintain institutional memory and manage this integrated system effectively are particularly relevant.

Additionally, scholars should investigate how polycentric systems, which are often praised as a solution for complex water management, may actually produce maladaptive outcomes in the presence of poor coordination, power asymmetries, a lack of leadership, disincentives for proactive change, and more (Biddle and Baehler, 2019; Lubell et al., 2014; Morrison et al., 2019). Empirical research on these processes would give important insights.





4) Research needs to be developed to better understand the role of drought pre-conditions and post-drought recovery in different systems. These would need to take into account dynamic vulnerability (de Ruiter and van Loon, 2022) and the interaction between long-term changes and short-term dynamics in different components of the systems. Long-term changes can include climate change, ecosystem composition changes, socio-economic changes and changes in land and water use / management, which all influence catchment, ecosystem and social memories. From the analysis of these short- and long-term dynamics, the occurrence of types and type transitions can be inferred. It would be informative to investigate under what conditions these types and type transitions occur. For this, we suggest analysing contrasting cases in different parts of the world, by combining observational data with modelling.

5) More research is needed on interactions and feedback between systems related to drought impacts and responses. For example, studies on the interactions between drought and wildfires should not only include ecological processes, but also hydro-climatic and social processes. Similarly, groundwater depletion should be analysed using approaches that include the complexity of hydro-social interactions over time (Schipanski et al., 2023). Also studies on maladaptation to drought should take into account the interactions and feedback within the hydro-eco-social system (Adla et al., 2023). Furthermore, we advocate for more collaboration between climate scientists and ecologists (Mahecha et al., 2022).

6) Tools for analysing drought as a continuum need to better accommodate interactions between systems and shifts in types. This could, for example, be done by combining the analysis of historical causal pathways (Srinivasan et al., 2012) with the development of future adaptation pathways (Haasnoot et al., 2013). A promising approach is that of storylines, which have recently been used to look at the climatological processes underlying drought (Gessner et al., 2022; Shepherd et al., 2018; van der Wiel et al., 2021). Storylines can also be developed from hydrological and ecological data (Bastos et al., 2023), and be combined with qualitative social narratives to show the unfolding of the past or of plausible futures of the interconnected hydro-eco-social system. Also earth system models, agent-based models and system dynamics models are tools that explicitly allow for these interactions to be explored (see example in Bastos et al., 2023; de Ruiter and van Loon, 2022).

## 5.1 Practice outlook

Dealing with drought as a continuum in practice will require changes to how droughts and their impacts are monitored, modelled, forecasted and managed. We suggest that:

1) Drought monitoring needs to move from an event-based to a continuous monitoring, for both hazard, vulnerability, and impacts. While several drought observatories still consider droughts as single events or are only operational in specific seasons, there are some ongoing efforts to move to continuous monitoring that can serve as an example. For drought hazard, the European Drought Observatory (EDO, 2023), East Africa Drought Watch (EADW, 2023), and Rijkswaterstaat (see 'Rhine River Basin' case study; Appendix 5) are moving towards continuous monitoring. For social and ecological drought impacts (Martínez-Vilalta and Lloret, 2016), the Drought Management Centre for



Southeastern Europe (DMCSEE, 2023), Kenya National Drought Management Authority (NDMA, 2023), and Brazilian Drought Monitor (Walker et al., 2024) (see 'Northeast Brazil' case study; Appendix 3) are examples where impacts are monitored on a continuous basis. Impacts on key ecological functions (plant productivity, water use, etc) are monitored continuously and through multiple remote-sensing platforms and ecosystem monitoring networks with global or regional coverage (ICOS, AMERIFLUX, FLUXNET, LTER). We are not aware of examples where vulnerability is monitored dynamically. We therefore recommend that key drought vulnerability indicators should also be monitored dynamically.

2) Monitoring of different systems needs to be combined to provide an overview of cascading effects between systems. The US Drought Monitor (USDM, 2023) is an example of combined drought hazard and impact monitoring that incorporates memory effects in different systems. The weekly drought map is a combination of physical drought indicators, drought impacts, field observations and local insight from a network of more than 450 experts, including hydro-climatologists, ecologists, forest scientists and relevant stakeholders. The maps are based on the information of the previous week and updated with new information. This approach builds in memory effects and explicitly includes drought recovery, both in the hazard and in the impacts. What is not explicitly included, however, are vegetation responses, for example changes in transpiration, and management responses, such as increased irrigation, which can drive changes of the hydro-eco-social system in time. We suggest other drought monitors to also include these memory effects in different systems and go one step further by also incorporating dynamic feedback.

3) Drought forecasting should be based on improved model tools that include memory and dynamic feedback (see Scientific outlook). Operational drought forecasting is currently limited to monthly precipitation and temperature forecasts, e.g., USDM drought outlook (USDM, 2023) and Latin American drought forecast (SISSA, 2023), excluding memories in the hydrological system. Also forecasting of drought impacts should be further developed to better anticipate societal and ecological impacts of drought. Some recent papers are taking steps in this direction, including longer-term processes, dynamic vulnerability and memory effects into impact forecasting (Boult et al., 2022; Busker et al., 2023). Operational drought impact forecasting is, however, still very limited (Shyrokaya et al., 2023). Only the East Africa Hazard Watch (EADW, 2023) includes forage forecasts and is in the process of developing food and water security forecasts.

4) Drought management should be more prospective. Prospective management means that, instead of only proactively reducing risk for an upcoming event, exposure and vulnerability are reduced long term and maladaptation and the creation of new risks are avoided (UNDDR, 2021). Drought management will always need to include an element of short-term 'early action' and crisis management to minimise unexpected impacts. But we argue that more attention should be paid to long-term adaptation and resilience building to avoid drought impacts and plan for the best strategies to reduce cascading effects within the hydro-eco-social system. Some recent approaches advocate for long-term drought resilience building and planning, e.g. the Three Pillars approach suggested by the Integrated



Drought Management Programme (IDMP, 2023), the related Drought Toolbox of UNCCD, (2023) and the EPIC Response framework of the WorldBank (Browder et al., 2021). However, these efforts need to be accelerated and scaled up and many established drought policies following this long-term approach are not implemented.

5) Drought management should be more coordinated and integrated across actors and systems. The current approaches for governing drought, and water more generally often contribute to a loss of social memory and maladaptation. This is because drought management is often highly siloed across different ministries or agencies due to its widespread effects on nearly all aspects of society (Bressers et al., 2016). This leads to significant fragmentation in responsibility for managing drought across scales and sectors, which increases the complexity of governance (Teisman and Edelenbos, 2011). Consequently, calls for more collaborative and networked approaches to water management have become ubiquitous (Eberhard et al., 2017; Sabatier, 2005), though the implementation and effectiveness of such approaches are highly variable.

These recommendations to science and practice will, we hope, contribute to adopting a changed perspective where droughts are not considered as drought as a snapshot in time, but rather as a continuum of interrelated and dynamic hydro-eco-social processes. Considering drought as a continuum, will require a change in how droughts are monitored, modelled and managed, but will provide an opportunity for a more holistic and integrated approach to managing droughts and the impacts they have.

**Code/Data availability**

No data was used for the manuscript.

**Author contributions**

A.F. Van Loon, M. Werner, S. Kchouk, A. Matanó, J. Mård, G.G. Ribeiro Neto, R. Biella, S. Khatami, A. Shyrokaya and M.L.K. Wens developed the original paper idea. A.F. Van Loon, M. Werner, G. Vico, E.A. Koebele, S. Kchouk, A. Matanó further developed its scope and structure. A.F. Van Loon, S. Kchouk, A. Matanó, K.E.A. Hassaballah, F. Tootoonchi, M. Wu, A. Shyrokaya, M.L.K. Wens, and M. Werner performed the literature review and wrote the sections, with contributions from R. Biella, Y. Du, P. Trambauer, C. Alvarez-Garreton, J. Mård, C. Teutschbein, A. Bastos, F.T. de Vries, C.H. Luce, V. Nagavciuc, G. Vico, L. Cavalcante, E.A. Koebele, E. Ridolfi, V. Aich, R. Tootoonchi, and support from I.G. Pechlivanidis, T. Roy, M. Galleguillos, M.H. Barendrecht, J.K.L. Koehler, I.N. Streefkerk, R. Weesie, M.L.K. Wens, and R. Stefanski. C. Alvarez-Garreton, E. Ridolfi and A.F. Van Loon led the case study collection. J.P. Boisier, L. Cavalcante, M. Galleguillos, M. Garcia, R. Garreaud, M. Ionita, S. Kchouk, E.A. Koebele, M. Mwangi, G.G. Ribeiro Neto, I.N. Streefkerk, R. Weesie, and M. Werner developed and analysed the case studies. A. Matanó and A.F. Van Loon developed the figures, with input



and feedback from F.T. de Vries, S. Kchouk, M.L.K. Wens, M. Werner, M. Wu, L. Cavalcante and G. Vico. A.F. Van Loon wrote the original draft with support of A. Matanó, E.A. Koebele, F. Tootoonchi, R. Tootoonchi, G. Vico, M.L.K. Wens, M. Werner, M. Wu, and V. Nagavciuc. All co-authors revised and edited the paper.

**Competing interests**

795    At least one of the (co-)authors is a member of the editorial board of Natural Hazards and Earth System Sciences.

**Acknowledgements**

The idea for this paper was developed during the IAHS Panta Rhei Drought in the Anthropocene workshop in Uppsala (Sweden) in August 2022. Elena Mondino was involved in earlier discussions related to this paper.

**Funding sources**

800    A.F. Van Loon, A. Matano, R. Weesie, H. Mendoza and M. Barendrecht were funded by the European Union (ERC StG, PerfectSTORM, grant agreement No. 948601). S. Kchouk, L. Cavalcante and G. Ribeiro Neto were supported by the Dutch Research Council (NWO) and the Interdisciplinary Research and Education Fund (INREF) of Wageningen University, the Netherlands (grant no. W07.30318.016). M. Wu was supported by the Swedish National Space Agency SNSA (Dnr 2021-00111), the Swedish Research Council for Sustainable Development FORMAS (Dnr 2022-00643), and Stiftelsen Oscar och
805    Lili Lamms Minne (Dnr FO2022-0016). V. Nagavciuc was partially supported by a grant of the Ministry of Research, Innovation and Digitization, under the "Romania's National Recovery and Resilience Plan - Founded by EU - NextGenerationEU" program, project "Compound extreme events from a long-term perspective and their impact on forest growth dynamics (CExForD)" number 760074/23.05.2023, code 287/30.11.2022, within Pillar III, Component C9, Investment 8. A. Bastos was funded by the European Union (ERC StG, ForExD, grant agreement No. 101039567). M.
810    Garcia was supported by the National Science Foundation CAREER grant: Balancing Local and Systemic Resilience in the Western Water Network (CIS-1942370). S. Khatami was supported by the Swedish Mannerfelt fond and Ahlmanns fond. I. Pechlivanidis was funded by the EU Horizon Europe Project MedEWSa (Mediterranean and pan-European forecast and Early Warning System against natural hazards) under Grant Agreement 101121192. M. Werner and R. Biella were partially funded by the I-CISK project under the Horizon 2020 research and innovation programme (grant agreement No 101037293).
815    Views and opinions expressed are however those of the authors only and do not necessarily reflect those of the funding agencies (e.g. the European Union or the European Research Council). Neither the European Union nor the granting authority can be held responsible for them.



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

The megadrought has led to the drying out of emblematic water bodies, such as the Aculeo lake, mostly due to the combination of decreased surface inflows, decreased groundwater recharge that led to groundwater disconnection from the lake, and water extractions for human activities (Barría et al., 2021b). It remains unclear how long it will take for these new hydrological states to recover after the megadrought finishes. In short memory pluvial basins water supply is more strongly dependent on the meteorological conditions of the current year, and thus it can be expected that a wet year would lead to a faster recovery than in long-memory snow-dominated or groundwater-dominated basins, or where groundwater has been disconnected from shallower water during the megadrought. The challenge is that most of these long-memory catchments correspond to semi-arid basins where irrigated agriculture is concentrated.



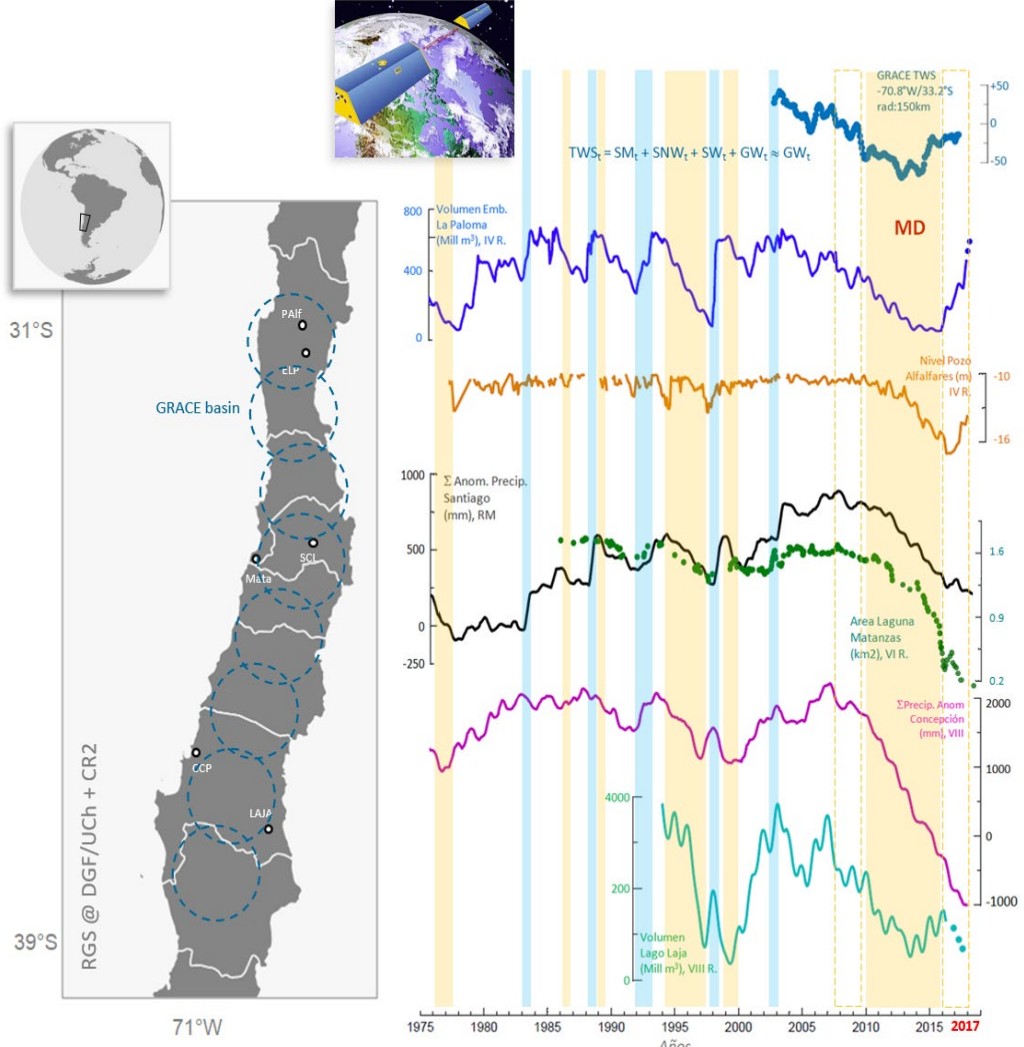

**Figure A1: The time series in the figure represent the response of different hydrological variables during the megadrought. The precipitation declines in central and southern Chile (black and magenta lines) have led to sustained decrease in the volume of the Laja lake (light blue) and Paloma reservoir (dark blue), and to almost full disappearance of the Matanzas lagoon (green dots). Underground declines are represented by the anomalies in total water storage from GRACE and by the water levels of the Alfalfares well.**

### Ecosystem

According to satellite observations, natural ecosystems have experienced significant vegetation browning following the extreme hyper-dry year of 2019 (Miranda et al., 2020). In the field, we found mortality of less drought-tolerant tree species, the complete drying out of several branches and even the trunk of more resilient species that have subsequently resprouted in




the year following the hyper-drought following a vegetative regeneration strategy. The current lack of seedlings depicts a risk of altering the composition and structure of these plant communities over time.

There is a relationship between drier conditions during spring and summer and a higher occurrence of wildfires, as well as a
larger burned area. Before the megadrought, wildfires were concentrated between the months of November and April, but now they extend from October to May, increasing the occurrence period from 6 to 8 months. Over 70% of the megafires have occurred between 2010 and 2018 in our country, where 50% of the burned area corresponds to industrial plantations. Notwithstanding this, it is worth noting that 99% of wildfires in Chile are caused by human actions, whether they are accidental or intentional. Therefore, the behavior of the population is crucial in preventing and mitigating wildfires
(González et al., 2018).

**Social system**

Some parts of the country have a precarious water infrastructure, particularly in rural areas, where there are issues of water access even during wet periods. In this context, the mega-drought has impacted a system that was already vulnerable before 2010, and these impacts have not diminished over time. On the contrary, they have intensified in some cases. This has
created an environment of increasing discomfort in rural communities, which currently lack adequate infrastructure to access water in sufficient quantity and quality. The general perception is that water management in Chile adopts reactive measures to face water scarcity in the short term, without an adequate plan for persistent drought conditions. This risk perception gets worse when climate change is also considered, given the drying precipitation trends projected for central Chile.

An important adaptive measure taken by governmental agencies to face water scarcity has been to build deep pumping wells
for supplying drinking water for human consumption. This measure, however, is causing groundwater depletions beyond those driven by the precipitation deficits and thus is not robust to face climate change. This implies covering permanent water uses relying on depleting underground savings, without considering that these savings are not being replenished.

Another measure has been supplying water to rural communities by cistern trucks, which is a non-structural reactive measure to address permanent water access requirements in rural communities (Nicolas-Chloé et al., 2022).
Social vulnerability varies across sectors. In rural areas, people rely on self-organized communities with inadequate infrastructure and technical capacities for providing subsistence drinking water, leading to water cuts that have been remedied by cistern trucks (Nicolas-Chloé et al., 2022). On the other hand, people in urban areas rely on water sanitation companies and have not been affected by water shortages, even when surface reservoirs have been significantly depleted since these companies have adequate groundwater infrastructure.
The evidence in rural areas has shown that reactive legal devices make drinking water precarious and can lead to overexploitation and contamination of water resources (Nicolas-Artero, 2022).

Furthermore, the slow adaptation of policies to a non-stationary hydrology and the drying trends observed in Chile makes the water management system inadequate to face droughts. For example, most of the current water use rights were allocated as absolute flows based on the water availability from decades ago and, in seeking to provide legal certainty, the law does not



permit to modify these allocated flows considering the current and projected climatic conditions (Barría et al., 2019, 2021a). This prohibition has led to overallocation of water use rights, as well as impeding the protection of environmental flows, and thus threatening the opportunity to reach water security (Alvarez-Garreton et al., 2023).

Despite the critical water deficits experienced during the megadrought, no water shortages have been applied to drinking water in urban areas since drinking water provided by sanitation companies mostly relies on deep groundwater reservoirs,
causing the disconnection between shallow and groundwater systems. In this context, some sectors are not experiencing the consequences of a decade-long megadrought, causing overreliance on a system that has critical vulnerabilities.

Climate projections for this region show a consistent decrease in precipitation. The drought impacts Chile is currently facing will exacerbate in the following decades. Water management measures should prepare for these projections, advancing from the reactive approach currently adopted.

**Management**

The Chile megadrought is framed in a climate change context, where formal attribution to anthropogenic warming has been carried out (Boisier et al., 2016, 2018), and is impacting a region where drying trends are consistently projected by GCMs. It is critical that actions are taken now. Water management recommendations to reduce drought-related risks include:

1) Adapting the water management system to account for a changing climate (Barría et al., 2019, 2021a)
2) Strengthening the protection of environmental flows to avoid water scarcity (Alvarez-Garreton et al., 2023)

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



Natural variability in the system may also impact actors' willingness to pursue adaptation to drought across sectors (e.g. a wet winter in 2023, following many years of drought, affected policy discussions in the Colorado River Basin around the stringency and timing of drought response measures). Similarly, highly partisan politics in the U.S., coupled with continued disbelief in climate change/science by some groups, may also lead some decision makers to "forget" or "ignore" drought impacts and expected future trends or delay adaptation (e.g. hesitance to raise water prices or invest in infrastructure adaptations, unwillingness to implement unpopular conservation measures in some areas, promotion of "natural variability" while ignoring broader climate and drought trends).

**Management**

The U.S. federal government has recently allocated historic amounts of funding for drought mitigation in the Colorado River Basin (2022-2023), recognizing the recurrence and "stacking" of drought impacts. However, there is concern that much of this money will be spent on temporary conservation measures (e.g. compensated fallowing of agricultural lands), which may not prepare the Basin for future droughts or a more arid future. Funding has also been allocated to help Indigenous communities address unsettled water rights and lack of infrastructure, and additional adaptation funding is available to some communities through state (e.g. California) and federal programs (e.g. WaterSMART). To be more resilient in the future, all states and sectors in the Basin must consider how to incorporate expectations of repeated and recurring droughts, against a background of broader aridification, into their response and adaptation actions.

Fortunately, water managers are increasingly incorporating observed and predicted climate change into their management efforts (e.g. urban utilities like Denver, Colorado; Phoenix, Arizona). Similarly, scientists and managers are working collaboratively to model Colorado River flows at different levels that represent a range of historical and predicted drought conditions. Increasing collaboration and the sharing of innovation across geographies and sectors within the basin can help support adaptation and reduce the negative consequences of drought, though such actions must continue even in wetter times to prepare for expected future climatic conditions.

Finally, governance processes must continue to become more inclusive of the wide variety of actors impacted by drought, including those who have been historically marginalized, such as Indigenous communities and the environment, to promote equitable adaptation and sustainability (Berggren, 2018; Koebele et al., 2023). There is a well-recognized need for increasing adaptiveness and flexibility in governance to deal with greater hydrologic variability and extremes and their impact on people and the environment.

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

**Ecosystem**

…

**Social system**

The vulnerability to drought and related impacts fluctuated over time in the semi-arid drought-prone rural community of Riacho da Cruz, in Northeast Brazil (Kchouk, 2024). This fluctuation of vulnerability resulted from the progression of a multi-year drought event coupled with drought responses. To address the drinking water insecurity in the community, a reservoir was introduced by national- and state-level water agencies. The rural population previously having their livelihood

based on manual workforce and subsistence farming started intensive livestock farming with forage irrigated from the dam. In the first years, the newly autonomous farmers decreased their vulnerability to drought by additionally relying on irrigation during the dry season to increase their income through high-value market livestock products. However, multiple years of drought and having their livelihood depend on this single activity made the farmers over-rely on the reservoir and irrigate the whole year; farmers' vulnerability to drought started increasing. Once the reservoir completely dried, the loss of production

coupled with buying forage from markets shocked by the drought, progressively depleted the finances of the farmers in the community. Ultimately, the lack of recovery and the prolongation of the drought lead to the collapse of the farmers' livelihoods.

In Brazil, the Northeast has a reputation of "a problem region, the poorest in the country, the most disadvantaged" (Théry, 2012). In addition to drought often invoked, the poverty and social vulnerability of the Northeast region are mostly linked to

the original latifundia system. The family farming system, representing nowadays 80% of the agriculture in Northeast Brazil but detaining only 37% of the agricultural lands (de Aquino et al., 2020), originates from a colonial law in 1850 that led to the division of large farms into small communities (Sabourin and Caron, 2001). With droughts making agricultural production uncertain, the small farming economy remained limited to meeting consumption needs. In addition, latifundists





restricted equal access to water by maintaining the reservoirs on their own lands. Successive divisions by inheritance led to the fragmentation of farms into strips, where plots are in length and aligned, to guarantee access, even limited, to water and the most fertile soils of the lowlands. This configuration turns collective management at the lowland or watershed scale particularly difficult and complicates the construction of water use infrastructure (e.g. receding, irrigation, access for herds, fences) (Sabourin and Caron, 2001).

The government's memory of solutions to drought events was that the impacts could be solved by increasing the water supply. The common practice to deal with drought events was to build large water infrastructure, such as dams and water basin transfers. This approach is known as the fight-against-drought paradigm. Over time, another paradigm gained prominence incrementally, the cope-with-drought paradigm, a proactive attitude toward nature, seeking to adapt to the environmental and climatic context. Both paradigms co-exist and compete within the governance system, however with dominance of fighting-against-drought (Cavalcante et al., 2022).

Drought is managed mainly with a reactive approach by drought commissions commanded by the Presidency at the federal level and by committees at the state level. These are temporary organizations, often criticized for not being able to respond quickly with comprehensive and integrated actions (Martins et al., 2016). A proactive approach started to be adopted with a policy instrument called Drought Monitor (Gutiérrez et al., 2014). In its most visible form, it is a monthly map that describes the current state of drought. However, more important than the map are the processes that encompass monthly meetings to
discuss the current drought conditions locally. This routine improves institutional and operational capacities to respond in an ongoing manner (Cavalcante et al., 2023).

The reactive approach to fight-against-drought resulted in two decades of reservoir building, strongly supported by the state (Silva, 2003). This fostered a safe development paradox with rural populations overly relying on reservoir storage for their income and livelihood (Campos, 2015). Conversely, the broad implementation of cisterns decreased this overreliance. These
water infrastructures of rainwater harvest at a household scale, alleviated farmers' dependence on reservoirs, ensured water- and food security and improved farmers' knowledge and confidence to deal with drought risks (Cavalcante et al., 2020; Mesquita and Cavalcante, 2021).

**Management**

There is a need to elaborate and integrate into existing DEWS, indicators that take into account the dynamism proper to
drought vulnerability. Such approaches can be based on the resilience of Social-Ecological Systems (SES), as it allows to understand a system's (in)ability to cope with and recover from drought events, as a result of always-evolving and dynamic biophysical and socio-cultural processes (Kchouk, 2024). Another possibility is to continuously monitor the drought impacts on affected populations, just like drought drivers are monitored and integrated into DEWS. In Northeast Brazil, drought impacts have been monitored by a network of local observers since 2019 (Walker et al., 2024).

Drought impacts monitoring is conducted on the ground in much of Brazil, since 2019, by local observers at monthly and municipality scale to support the Brazilian Drought Monitor. The open nature of the questionnaire means the programme is a



globally rare and consequently valuable example of drought impacts monitoring by the people "on-the-ground" who experience the impacts. Crucially, this type of regular spatially distributed monitoring should provide both baseline conditions and the effects of any disturbances (Walker et al., 2024).

The memory of past dry events is directly linked to hydrological aspects, such as the reduction of hydraulic connectivity due to the presence of a dense network of reservoirs, as well as social factors, such as the history of public policies to fight against/cope with drought. Therefore, it is necessary to consider these dynamics in the process of modeling drought impacts. Traditional approaches based only on the representation of the physical components of the hydrological cycle are not able to englobe this complexity, requiring multidisciplinary approaches, such as the application of socio-hydrological models

(Ribeiro Neto, 2024). Some studies in Ceará have been successful in this regard by using agent-based models (Van Oel et al., 2008, 2012).

Planning and integration among institutions have been one of the main challenges related to government responses to droughts in Brazil. The most recent aspect of governance has been the implementation of policy instruments aligned with the idea of drought preparedness, for instance, the Drought Monitor. This policy instrument is the first attempt to overcome the

challenges to proactive and integrated governance at distinct levels continuously, not only when the drought impacts are identified.

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

## Hydrological system

The Horn of Africa (HOA) region has high variability in climate and catchment characteristics, and therefore also how drought propagates. Odongo et al., (2023) found that arid and semi-arid areas in the region, propagation from meteorological to soil moisture drought is influenced by surface processes, such as soil properties, land cover and the time of the last rain – variables which are all linked to the storage capacity of the catchment. In areas where the soil is very dry, the soil surface needs to be wetted before infiltration starts. Therefore, catchments in the region with a high aridity and high sand content have a slow response of soil moisture to precipitation. In the HOA, the propagation from meteorological to streamflow drought is largely influenced by catchment-scale hydrogeological processes, such as geology and land cover.

Despite increasing frequency of droughts in the Horn of Africa, including Kenya, since the early 2000s, sustained water storage has increased. How is this possible? Heavy rainfalls in the Horn can play an important role in dampening subsequent drought duration and severity. High intensity rainfalls, especially during the OND (October-November-December) rainy season, lead to large seasonal increases in water storage that persist over multiple years. Therefore, increasingly recurrent drought damages could be mitigated with groundwater resources recharged by heavy rainfall events (Adloff et al., 2022).

## Ecosystem

In Kenya, recurrent droughts and rising temperatures cause more and more often food and water shortages for herbivores, resulting in an ecosystem response of greater movements and likelihood of contacts between wildlife, people and livestock. For example, in Narok County, during the severe drought years of 2008-2009, there were the highest recorded number of human-wildlife conflict. The conflicts resulted fromwild herbivores encroaching on cropfarms, or predators attacking livestock (Mukeka et al., 2019). Human-wildlife conflicts in Kenya often result in long term damages for both sides: crop farmers and livestock herders experience a reduction incapacity to recover from droughts and prepare for the next.. Wildlife often pay with their lives, especially nonhuman primates, which are often trapped and killed by farmers protecting their precious crops, usually out of sight of the authorities of the Kenya Wildlife Service (KWS).

Riparian forests in Kenya, often forming thickets, used to provide habitat for wildlife and act as an insurance against drought for both humans and wildlife. The strong decline in riparian forest area in Kenya because of agricultural encroachment have thus removed this ecological buffer against droughts in many areas (Schmitt et al., 2019). The ongoing removal of riparian forests is likely to speeden up and intensify drought impacts, and reduces the resilience of riverine ecosystems, including its human and nonhuman inhabitants, to recover from droughts and prepare for the next.



Also in mountaineous forests, in the Mt Kenya region, human activities in the form of agricultural intensification and the expansion of horticulture agribusinesses have increased pressure on water resources, pasture, and idle land because of a shrinkage in area of natural and agro-pastoral landscapes (Eckert et al., 2017).

**Social system**

In 20th century Kenya, droughts generally took place every 5 to 10 years and were usually considered, by both pastoralists and drought experts, as 'events' that one could recover from. In the last two decades however, more frequently recurring droughts (every 2-3 years) have impoverished pastoral communities and thereby changed the younger generation of pastoralist' perception of drought risk. In northern Kenya for example, the recurrent droughts in the last decade, the current

one having started in 2020, has disabled pastoralists to restock their livestock herds, and has forced them to reconsider their grazing management. Previous grazing management systems, based on 5-to 10-year droughts and therefore having long recovery times, have demonstrated not to be adaptive anymore in 21$^{st}$ century Kenya. As Thomas et al., (2020) state, because of the increase in drought frequency, there is no longer enough time for pastoralists to recover their herds and crops in between droughts, thus the destabilizing impact increases with each successive drought, leading to drought vulnerability.

Realising this, a new generation of pastoralists has come to see droughts as an ever-recurring state of affairs; one that requires a renewed, more strict graze zoning management to allow pastures to recover from recurring drought conditions (NRC, 2023b, 2023a). By doing so, they acknowledge the importance of land cover, and hence land use, in the propagation of recurrent drought.

In Southern Kenya, local authorities and NGOs have responded to consecutive droughts in the 1990s and 2000s by

constructing public dams, water reservoirs and irrigation systems for crop-farming in previously pastoral areas. These schemes, when implemented and managed locally, can help in improving resilience to droughts, by giving pastoralists an alternative source of income and food supply which is not solely dependent on livestock production. It has led to the intended resilience building, at least for those able to reap its benefits. The vulnerable however have ended up in poverty traps: now, they had to compete in their extensive pastoral land and water uses with even more cropfarmers, all of them

attracted to settle around the new water infrastructures (the infrastructure set in motion a divergence in adaptation trajectories (Weesie and García, 2018). Besides, these types of irrigation schemes in sparsely populated areas tend to induce growth of people and livestock numbers, who all rely on the newly available, publicly accessible, water. It puts not only more pressure on the water source to perform, but also on surrounding natural resources, such as pastureland and seasonal rivers. Hence the schemes, while having short term benefits, on the long run induce an overreliance on reservoir storage, leading to a

landscape with a higher likelihood of even heavier drought impacts in the future (Weesie and García, 2018).

In the Horn of Africa, historically, policy responses to drought have been reactive. Change was promised after the heavy 2010-11 drought in Kenya, when the Kenyan National Drought Management Authority (NDMA) formulated several key response activities during drought emergencies: a) maintenance of groundwater boreholes, b) installation of temporary 'dry-season' boreholes c) water trucking of purified drinking water to affected communities. (GoK 2015, in (Thomas et al.,



2020)). While such efforts can be praised, it reveals how policy responses still are of a rather ad-hoc and reactive nature, with institutions only slowly learning after heavy droughts.

The Horn of Africa is currently facing high food insecurity, affecting millions of people. Several failed rainy seasons have caused the 2020-23 drought period to be one of the worst in recent decades. However, drought hazard is not the only driver of food insecurity. Food insecurity and systems are complex and have many other drivers than climatic ones (Sandstrom and

Juhola, 2017). The current food insecurity situation in the HOA is compounded by instability, conflict, the impact of the COVID-19 pandemic and rising food prices (WHO, 2023). The understanding of these food insecurity systems is advancing, however, there does not seem to be a change in the humanitarian responses. Short-term emergency relief remains evident (common practice?), partly due to the 'blaming of the rain', instead of moving towards more advanced approaches that consider other drivers of food insecurity (Sandstrom and Juhola, 2017).

**Management**

- In 2016, the Kenya National Drought Management Authority (NDMA, 2023) changed their assessment from an event-based approach to continuous drought impact monitoring. They now produce a monthly national drought early warning bulletin to coordinate drought risk management and to establish measures to mitigate drought emergencies in Kenya, either on their own or in collaboration with stakeholders.

- The East Africa Drought Watch (2023) uses similar indicators as EDO to monitor drought hazard conditions in the East African region, but allows for mapping these indicators on different timescales from 10 days and monthly, to seasonal and yearly.

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



conditions rapidly returned to a drought state in 2022, raising questions about whether changes in recharge rates or rainfall-runoff processes have occurred.

One of the most notable impacts of the 2022 drought was on the Rhine River, Europe's most important inland waterway. The low water levels in the Rhine disrupted shipping, threatened biodiversity, and highlighted the vulnerability of the river to climate change. In August, the streamflow at Köln, Germany, reached a historic low of 652 m3/s. Over most of the month of August, the water levels reached unprecedented low levels over the last 140 years (Fig. A2). Morover, on the 17th of August 2022, due to the extreme ongoing drought, the water level of the Rhine in Emmerich, near the Dutch border, had reached a historic low of -3 cm. These low water levels were caused by a combination of factors, including: reduced rainfall, high
temperatures and groundwater depletion. The prolonged drought also led to depletion of groundwater reserves, which are a major source of water for the Rhine and other rivers.

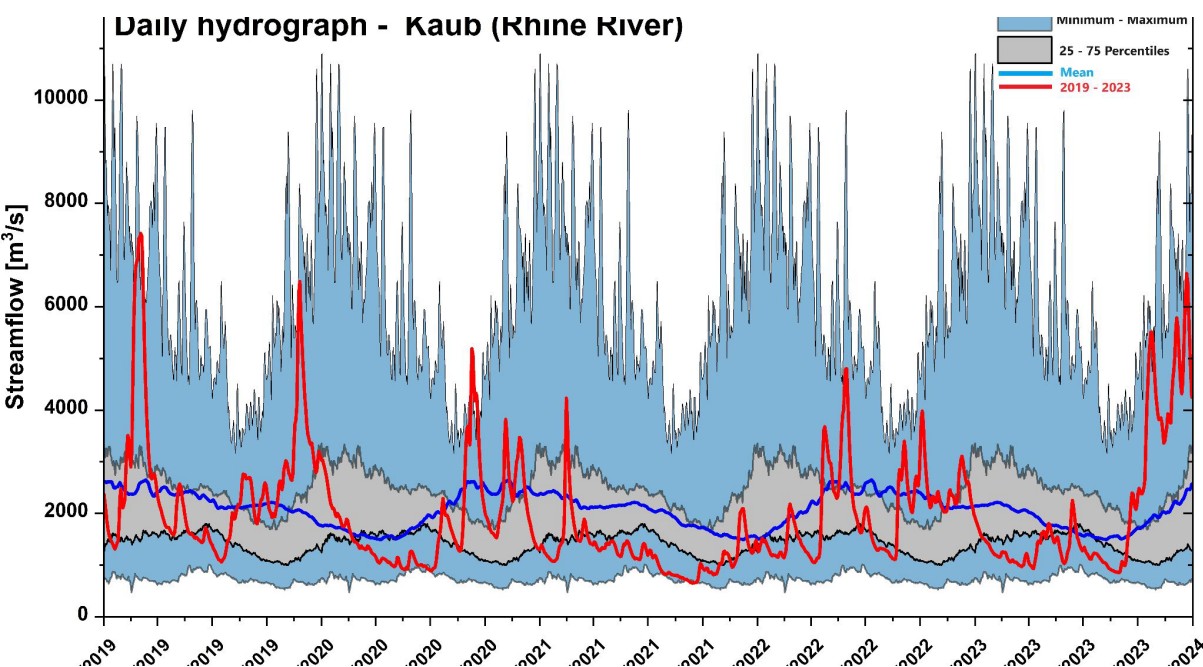

**Figure A2: Daily hydrograph at Köln gauging station (Rhine River) over the period 1.01.2019 - 31.12.2023. The period 1881–2018**
**was used to compute the daily streamflow climatology.**

**Ecosystem**

The relationship between anomalies in soil moisture and vegetation in the Rhine basin is complex and soil moisture droughts do not always correspond directly to negative vegetation anomalies (Van Hateren et al., 2019). In some years, drought conditions result in a positive response of the vegetation, in other years negative vegetation anomalies occur in periods that
are not extremely dry, probably because other factors play a role as well. The role of antecedent conditions seems crucial, especially in spring, and only later in the year the relation between soil moisture and vegetation anomalies becomes more





synchronous (Van Hateren et al., 2019). But a wet spring does not always result in a higher resilience to drought later in the year. In the Netherlands, the winter and spring of 2021-22 showed wet pre-conditions for the drought that developed in summer 2022. However, the extensive drainage system that historically was used to discharge any excess water quickly also meant that the water was not stored and that starting conditions of the hydrological system were sub-optimal, so that the dry summer could result in drying out of local streams and ponds and impacts on aquatic ecosystems (Natuurmonumenten, 2022).

During the recent droughts in Europe (2018-2020) several legacy effects became apparent in ecosystems in the Rhine basin. Buras et al., (2020) showed that pastures and arable land were more vulnerable to variabilities in the climatic water balance and had a stronger immediate response to drought than forests. These differences are largely associated with the forest-grass differences in rooting depth-related hydroclimate conditions. In some cases, forests tend to have larger rooting depth (Schenk and Jackson, 2002) and larger root-zone water storage capacity to buffer severe droughts (Mackay et al., 2020; Wang-Erlandsson et al., 2016). But the two consecutive hot and dry summers in 2018 and 2019 amplified impacts due to preconditioning from past disturbance legacies (Bastos et al., 2021). The 2018 drought event severely impaired the physiological recovery of trees, making them vulnerable to secondary impacts, such as insect or fungal attacks, with the resulting tree mortality expected to persist for several years. As a consequence, tree mortality rates in Germany spiked in 2020, at values about 10 times higher than in the past decade for needle-leaved trees (BMEL, 2021) and this is likely to continue for several years. Also several ecosystems and specific species in the Netherlands were affected by the accumulated effects of several dry years in a row, combined with other pressures such as pollution (Natuurmonumenten, 2022). This is an example of how ecosystem effects of drought can be creeping and accumulating (Tijdeman et al., 2022).

**Social system**

Most drought impacts in the Rhine basin were recorded in the years 2003, 2015, and 2018. (Dahlmann et al., 2022) found that these were not evenly spread over the basin, but that more impacts were recorded in downstream parts of the basin, confirming the hypothesis of asymmetric upstream-downstream impacts.

Drought impacts are not always felt by society in the Rhine basin. This is partly because baseline vulnerability is low, partly because effects are compensated economically (for example by higher market prices resulting in a net-zero effect of drought on agricultural or shipping revenue), partly because impacts are passed on in time or passed down to other systems. This passing on of impacts for example happened in 2018-19 in the Netherlands when increased groundwater abstraction for irrigation was increasingly used as adaptation strategy after the 2003 drought (Kreibich et al., 2022a). This reduced the drought impacts in irrigated agriculture, but increased impacts in non-irrigated agriculture and groundwater-dependent ecosystems due to lower levels during the drought. It can also prevent recovery and create more impacts during subsequent drought events (Kreibich et al., 2022a).

A study on the governance of droughts and floods showed that in the Netherlands severe drought events (1976, 2003) resulted in less structural measures than flood events (1953, 1993; Bartholomeus et al. (2023). For example, in the



2235 Netherlands drought management is crisis management. Also, during drought often only restrictions on surface water abstraction are implemented, not on groundwater abstraction, which neglects the connections between these parts of the hydrological system and ignores the longer-term impacts that increased groundwater abstraction can have. Using the case study of adaptive water management in the Netherlands, Pot et al. (2023) demonstrate how policy actors can use temporal strategies to navigate dual crises like creeping and acute threats unfoling at the same time, such as extreme weather evens

2240 such as drought and flooding and the creeping crisis of climate change. These strategies include strategic coupling of long-term shocks and creeping crises, crafting time horizons, molding the pace of public problem-solving, inter alia.

Climate change makes systems that gradually adapted to a certain state, suddenly not adaptive anymore to the new normal in the future. This is what is currently happening in the Netherlands, a country very well adapted to flooding, but now needing a transformative shift to drought management. Also in other countries in the Rhine basin (Germany, France) and the rest of

2245 Europe a change in drought management is happening. For a long time the 1976 drought was the benchmark drought for policy, but only in response to the 2018-2020 drought new policies have been implemented (e.g. Blue Deal in Belgium, Bodem & water sturend in the Netherlands). This shows that the recent droughts have increased awareness.

### Management

The different interfacing signals (nival/pluvial, groundwater etc) implies the importance of tracking non-drought conditions

2250 (approaching drought) in a drought management context, especially for the more slowly responding rivers and aquifers. Until recently, the Royal Netherlands Meteorological Institute (KNMI) in the Netherlands only monitored drought conditions during the growing season (1 April - 30 September) and the effects of a wet winter were not taken into account (KMNI, 2023). KNMI recently also added continuous SPI monitoring, which the Ministry of Water now includes in their continuous monitoring bulletins of various hydrometeorological variables, including river discharge and groundwater (Rijkswaterstaat,

2255 2023). Nevertheless, the monitoring remains focused on the climatic water balance (precipitation minus potential evapotranspiration) (KMNI, 2023).

The so-called "drought radar" (Deltares, 2023) goes a step further to forecast drought conditions in groundwater. They even include an estimate of water management and groundwater abstraction, but this information is incomplete and not dynamic (Berendrecht et al., 2011). It is rare for these systems to operationally include hydrological drought and dynamics and

2260 feedbacks of ecosystems and social systems.

In terms of drought management, increasingly long-term pro-active measures are being implemented in the Rhine basin. For example there are many projects to increase infiltration and decrease drainage, e.g. by reducing drainage density, managed aquifer recharge, etc (Kreibich et al., 2022b, 2022a; Sprenger et al., 2017). In the Netherlands, water boards implement surface water use restrictions during drought, but after multi-year drought there is more awareness that groundwater use

2265 should also be restricted with the aim to prevent long-term effects in the hydrological system and potential cascading effects on the social system and ecosystem (Bartholomeus et al., 2023). In Germany, there are different projects which are trying to use agile network control to increase the resilience of water supply infrastructure, by developing a situation-dependent



customer (group) specific regulation of water quantities using AI technology, as well as a impelement a (pre-)operation low flow forecasting system for the water levels of Rhine.

There is also discussion on the need of a common drought management strategy within the transboundary Rhine basin, given the different degree of development of drought management and water allocation policies across the riparian states (Blauhut et al., 2022; Dahlmann et al., 2022).

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
