# Peer review of "REVIEW ARTICLE: Drought as a continuum: memory effects in interlinked hydrological, ecological, and social systems"

_EGUsphere, 2024_

## Author Response (AR1)

**Rebuttal (bold) to editor comment Floris van Ogtrop (normal text)**

Dear Anne van Loon and colleagues, thanks for provided clear and extensive rebuttals to the reviewers comments and the community comment. I have contacted both reviewers and they are happy with your responses to their comments, as am I. Please go ahead and implement the changes as outlined in your rebuttal and upload the edited version of your article. I will review the final version before final publication.

**>> Thanks for your positive evaluation of our rebuttal. We have implemented all suggested changes. See the detailed rebuttal below and the tracked-changes manuscript attached. We also fixed a few typos and other textual issues. We are looking forward to receiving your evaluation of our revised manuscript.**

**Rebuttal (bold) to review Rene Orth (normal text)**

This review paper summarises the state of the art of the research on drought impacts on hydrology, ecology and society. The authors advocate for viewing droughts as multi-spheres phenomena, and to move away from event-based drought analyses. They conduct five case studies of drought impacts in different regions of the world to illustrate specific drivers and implications, and use this to showcase a separation of droughts into four archetypes with distinct temporal dynamics.
* * *
Recommendation:

I think the paper requires moderate revisions.

The topic of this review is interesting and timely, and it is a great fit for the readership of NHESS.

**>> Thanks for your positive evaluation of our manuscript. We are happy to hear that you found it interesting.**

It serves as a useful summary of the state of the art of drought impacts across individual spheres, and at the same time it adds a new perspective by focusing on the complex nature and implications of droughts across geo- and socio-spheres. I like that this review is motivating more comprehensive research on droughts, which takes into account multiple spheres and can provide more profound insights. This can help to steer the very active research in this field. Consequently, research can then probably also yield more effective and sustainable management options. Benefits of this interdisciplinary research approach are convincingly illustrated with the case studies that include detailed descriptions and comparisons of common drought response strategies.

**>> We agree that the paper can inspire both research and management. Thanks for pointing out the benefits of our work.**

However, before the paper is ready for publication I would recommend to address a few discussion points:
* * *
General comments:

**>> Please find below our responses to your comments. Line numbers refer to the tracked-changes version of the manuscript.**

(1)

I like the consideration of the concept of drought memory across time and spheres. But currently the term memory is used in a somewhat confusing way as it is (i) used with and without "" ("memory" and memory) while the difference is not clear, and (ii) only defined in line 165 after it has been used several times. Another point in this context is that the paper focuses on development and implications of memory, but not so much on its dissipation through e.g. meteorological variability which can shorten memory more or less strong.

**>> Thanks for highlighting this inconsistency. In the revised paper, we now define the word memory earlier in line 81-82 (before its first use) and removed the "" in the rest of the text. We also added a few lines defining the terms memory, response and legacy and their relationship (l.197-204).**

**>> About the dissipation of memory, we discussed this mostly in the social system (Section 3.3), where we talk about recovery, how memory fades over time, a low memory of creeping events like drought, and forgetting drought. In the hydrological system (Section 3.1), we discussed the non-stationary of catchment memory due to extreme climate conditions and specifically mentioned decreases in memory due to changes in snow and glacier dynamics and human activities. In the ecosystem system (Section 3.2), we also talked about how the drought-modulated ecosystems feed back to the hydrological system and affect drought itself and how post-drought ecosystem responses can change. But we followed the reviewer suggestion and now discuss the dissipation of memory more explicitly in the revised manuscript in Section 3.1 (l.239, 248-250, and 297) .**

(2)

The outlook and recommendations section provides useful advice to the community. One aspect that I am still missing is some discussion or even guidance on how we could promote more interdisciplinary research approaching drought as a hydro-eco-social continuum. Given the focus and arguments of this paper, this could be another important contribution it could make. I can think of two directions in this context:

— Observational data: This is critically needed for enhancing process understanding and for constraining more sophisticated models. Following the systems approach of the paper, also data needs will change. For example social system impacts and characteristics need to be quantified in addition to (more readily available?) data on hydrological and ecological systems, and using such data streams from different spheres together might pose challenges with respect to spatial and temporal resolution and coverage of the data. It would be great if the authors could add some thoughts or even recommendations in this direction.

— Spatial and temporal scales: An issue which becomes apparent from the comparison of hydrological, ecological and social aspects of drought research are the different spatial and temporal scales considered within, but even more across these spheres. This raises the question how to determine adequate scales, and to which extent we can or should ensure to aim for comparable scales in order to facilitate interdisciplinary drought research across spheres. See for example De Polt et al. 2023.

**>> Thanks for this suggestion. We agree that observational data for interdisciplinary research on the hydro-eco-social drought continuum could be improved. We addressed this in the previous version only from a practice perspective in Section 5.2 (point 1 and 2 on monitoring). In the revised manuscript we added a point referring to this also in the research perspective (Section 5.1, point 7, l.789-796). We specifically mention that data need to be collected already with multidisciplinarity studies in mind.**

**>> We also agree with your suggestion on scales. In this paper, we mostly focus on temporal scales, and spatial scales are not addressed explicitly. Discussing spatial scales relevant for each system and for the drought continuum would require an in-depth discussion of hydrological, ecological and social processes that we feel is beyond the scope of the current paper. Nevertheless, we mentioned this aspect in Section 5 in the revised manuscript as a suggestion for future research (Section 5.1, point 8, l.797-801). We also added the reference suggested by the reviewer in the Introduction (l.128).**

(3)

Related to the drought trajectory types, in my understanding type 4 is similar to type 1b such that I do not see the necessity for a fourth category. Also, I feel that the addition of "big shock" in the name of type 4 is not needed as this may not necessarily be the case, which can also be seen from the case study examples.

**>> We see your point that the High-resilience, big shock category could be regarded as an Impact & recovery type with a very slow response. However, here we want to highlight that systems which seem to have a very high resilience, not reflecting the immediate effects of drought, could be building up underlying vulnerabilities that could bring the system to an unexpected tipping point. In the case studies we see a high potential for this type of dynamics in the Rhine basin and Chile (and a similar process happened in the Cape Town Day Zero drought), although we agree that in all these cases a collapse of the system did not happen (yet). We have considered to rename this type into "Tipping point (potential)", but we decided against it as that term does not include the initial high resilience. Instead we have now included some comments on critical transitions / tipping points (l.598-603) and adapted Figure 4 to show the tipping point behaviour better (to also be more in line with for example Figure 1).**

I do not wish to remain anonymous - Rene Orth.
* * *
Specific comments:

lines 53, 73, and others: Maybe some discussion can be added on the reasons for adopting an event-based perspective on droughts in past and present research. I think for example that some form of quantification of drought time periods (even though they may be different across spheres) is necessary for separating droughts from hydrological variability.

**>> Thanks for this suggestion. We added a discussion on the reasons for an event-based perspective on droughts in the past in the Introduction (l.124-148). Indeed, for extreme-value statistics drought event characteristics are needed and for drought management a disaster risk framing can be useful, which requires a hazard event to be distinguished with a specific exposure and vulnerability related to the hazard event.**

line 90: Here you could also cite O & Park 2024

**>> Thanks. We added the reference.**

line 139: "interacting systems interacting" is a bit too interactive I think

**>> Thanks. We removed the first interacting.**

lines 148-157: I like the mention to the concept of socio-ecological systems and Earth system science. However, it could be more integrated and compared with the drought continuum concept proposed here.

**>> We agree. In the paper we use this concept as the basis for our ideas and discussion. We did consider adding a Discussion section where we would come back to these concepts and elaborate on how the hydro-eco-social drought continuum is related to these, but we decided against this as it would make the paper longer since it is already almost 33 pages. In the revised version we added an extra sentence in the Introduction, to introduce these concepts from the start (l.156-162).**

line 285: Here you could also cite Li et al. 2023

**>> Thanks. We added the reference.**

line 294: would remove "depleting" here such that the previous "reducing" applies for the decomposition argument

**>> We removed "depleting".**

line 392: add "on rivers" after "reductions in goods transported"

**>> We added "via rivers".**

lines 375, 393, and others: it is interesting to see the similarities in the drought response of social system to that of ecological systems

**>> We agree. We made this more explicit in the revised version, e.g. l.426-428, 433-434, 443-444, 486-487, 517-519.**

line 718: should be section 5.2

**>> Indeed. Thanks for pointing this out.**

Table 1, bottom right box: Maybe replace "prevent" with "hinder"

**>> We replaced "prevent" with "hinder".**

References:

De Polt, K., et al., Quantifying impact-relevant heatwave durations, Env. Res. Lett. 18, 104005 (2023).

Li, W., et al., Widespread and complex drought effects on vegetation physiology inferred from space, Nat. Comms., 14, 4640 (2023).

O, S. and S.K. Park, Global ecosystem responses to flash droughts are modulated by background climate and vegetation conditions, Communications Earth & Environment, 5, 88 (2024).

**Rebuttal (bold) to review Ana Iglesias (normal text)**

The review article is interesting and well written, supporting the well-established call for a systems perspective to successfully manage all natural and man-made hazards, including adaptation to climate change. The article is a very major effort from the earth sciences community to engage with social sciences, and in this sense, its value is great. Section 3 of the manuscript is extremely well written and provides very interesting information.

**>> Thanks for your positive evaluation of our manuscript. We are happy to hear that you see the value in our work and find it well-written.**

However, the current version of the paper does not completely respond to the anticipated goal. Some changes may be interesting to further engage all readers, to provide an accurate state of the art, and to make progress to decrease the social damage by drought.

**>> Thanks for your suggestions. We incorporated them in our revised manuscript (see below). Line numbers refer to the tracked-changes version of the manuscript.**

Some minor comments:

1. Recognise the immense systemic contribution made in the developing of drought management plans.

**>> This is a good point. We added this to the Introduction (l.129-148, 150-151) and Section 3.3 (l.525-529).**

2. Further develop and discuss the idea of "memories" that is a well-known determinant to human responses to all adverse events. This could be included in the drought preconditions and recovery in different systems.

**>> We agree. Memory is one of the core aspects of our paper. We mentioned it extensively in Section 3.1. In the revised version of the paper, we define memory already in the Introduction (l.81-83), explain memory more extensively in Section 2 (l.197-204), changed the title of Section 2 to include memory, and added it more prominently also in Section 3.2 and 3.3 (e.g. l.331, 356, 469, 474).**

3. The idea of dynamic vulnerability and maladaptation is extremely interesting, but not completely developed in the paper.

**>> We agree that it is an interesting topic. Please note that we already discuss maladaptation quite extensively in Section 3.3, in the case study examples, and in the research and practice outlooks. Dynamic vulnerability underlies much of Section 3.3 and is mentioned in the research and practice outlooks. In the revised manuscript, we brought dynamic vulnerability and maladaptation more to the foreground in Section 3.3 (e.g. l.472-473, 490).**

4. In my view, trendy terms such as "flash droughts" of "mega-droughts" do not add and substantial concept and detract from the logic of the paper. But I do not object to their use if the authors really like them.

**>> We are not so much attached to the terms "flash droughts" and "mega-droughts". We want to keep them in Section 1, because we want to acknowledge that these are much-used terms in the literature  and used to depict temporal aspects of drought events and refer to some of the recent**

**studies on these. But we agree that in the rest of the paper these terms are not needed, so we removed them or replaced them by more neutral terms like multi-year drought.**

5. Eliminate the statements such as "we argue that understanding drought requires taking into account not only physical (hydro-meteorological) processes, but also ecological (environmental) and social (economical, political) processes to assess drought risks" that are not original. These very well-established ideas need to change "we argue" to "we support". (this is only one example; more similar statements are throughout the text).

**>> Thank you. We carefully re-read the text and rewrote these type of unoriginal statements (e.g. l.154, 843).**

6. Section 2 is the weakest section and needs to be re-written completely. Table 1 is extremely limited.

**>> Thanks for pointing this out. However, without more detailed comments it is difficult to know why the reviewer finds this section to be the weakest or what is needed to improve it. In Section 2, we wanted to give an overview of the concepts that are behind our analysis and to justify our focus and approach. Table 1 was not aimed to be complete but included to show that "time characteristics have been studied empirically in the separate systems (see some examples in Table 1)." and to justify that we are looking at these three systems (and not at other potentially relevant systems). We indicated this by mentioning in the caption of Table 1 that these are "examples" and "based on specific studies".**

**Nevertheless, in the revised manuscript we critically revised Section 2, including a better integration of the aspect of memory (also in response to comment 2, l.197-206) and changed the title of the section.**

7. Section 3 is the strongest section, especially Section 3.1. Minor changes in Sections 3.2 and 3.3 would bring them to the excellent level of Section 3.1.

**>> Thanks for the compliments for Section 3.1. We worked on Sections 3.2 and 3.3 and improved them in line with the reviewer's earlier suggestions**

8. Section 4 could be more explicit the application of the framework to the case studies.

**>> Thanks for the suggestion. In Section 4, we relate back to and illustrate the concepts of Section 2. We developed the four archetypical temporal drought trajectories, which are then applied to the case studies (Section 4.3). This is clearly introduced at the beginning of the section 4.3: " We explored different drought typologies, system-interactions and type-transitions in five case studies...". We would require more specific feedback to understand what to change in Section 4.**

9. Section 5 seems to be judgemental and with limited applications for drought management practitioners. A shorter section, more related to the on-going policy development could be more useful.

**>> Section 5.2 on Practice outlook is written by drought management practitioners (e.g. from ICPAC, IDMP) and provides both successful examples and suggestions for improvement. We agree that we could go more in detail on the practical applications, but considered that this was beyond the scope of the current scientific paper and could better be addressed in other ways (policy briefs, etc). A section on ongoing policy developments would be interesting too, but we feel that this would be too different from the message and arguments of the rest of the paper. We have set up Section 5 such that it links back to what was discussed in Section 3 to have a strong link with our**

**literature review and the suggestions. However, for more clarity, we added a paragraph to the manuscript (l.859-863).**

> 10. Finally, as many papers written mainly by non-social scientists, the link to social sciences seems to be the inclusion of extraordinary social scientist in the reference list. Here is the case of E. Ostrom. In my view, dropping names is rarely successful. If the authors want to recognise concrete contributions or linkages to their work, they should be more concrete.

**>> We disagree that this paper is written mainly by non-social scientists and we want to stress that our paper has two axes of interdisciplinarity, i.e. between hydrology and social science, and between hydrology and ecology. We had all authors self-identify to one or more of these research fields. Most authors work at the interface between hydrology and social science or between hydrology and ecology (24 authors self-identifying as interdisciplinary scientist). For the specific disciplines, we are biased towards hydrologists (20), but have a good team of ecologists (6) and social scientists (5) on board. Our inclusion of certain references is not meant as name dropping and it is unfortunate if it appears this way. Beside the reference to the work of Ostom in relation SES, we also use a lot of different social science literature, both in Section 2 and 3.3. We would be happy to hear from the reviewer more specific suggestions for including other social science literature.**

**Response (bold) to comment Carolina Ojeda Leal (normal text)**

In this article, more than 40 well-recognized authors proposed a new shift in the studies of drought phenomena. They took 5 cases from different parts of the world and applied an adapted version of system theories (resilience, SES, human-nature coupled systems, and collapse) to argue that drought is a long-term phenomenon that damages multiple systems (hydrological, ecological, and social). Overall, it is a very well-written article, with abundant literature references and the scientific rigor expected in a literature review made by 42 people. Nevertheless, it is so extensive and theoretically dense that sometimes looks more like a book chapter or white paper rather than a scientific article of NHESS, which in my impression intended to be more applied.

**>> Thanks for adding these positive comments to our manuscript. Your suggestions are very much appreciated.**

I have a few comments to improve the article:

1. In the section "4.3.2 Systems Influencing Each Other" paragraph 590 I suggest changing the term "industrial tree plantations" to "exotic species monocultures (e.g., pine, eucalyptus)". Also, I suggest citing CONAF in those statistics of data about wildfires resulting from their excellent work. Moreover, I do not understand from that paragraph if the "megadrought" in Chile and the social system (anthropical fire ignition + exotic species monocultures) by themselves instigated changes in the hydrological system which was translated into the extension of fire season. It was confusing to read sometimes because created the illusion that both local drivers ("megadrought" + social system) without global climate change could alter the Chilean fire seasonality, but it could be because the only data available to analyze the wildfire damage is from CONAF which started to keep it from 1985 until today. To establish more clearly that hypothesis it could be necessary to expand the four-decade analysis to incorporate the memory of pre "megadrought" time with appropriate literature if is relevant to the topic.

**>> Thanks for these suggestions.**

We changed the phrase: "*More than 70% of the megafires (>50,000 ha of burned area) of the last four decades have occurred during the megadrought, where 50% of the burned area corresponds to industrial tree plantations.*" To "*More than 70% of the megafires (>50,000 ha of burned area) of the last four decades have occurred during the drought, where 50% of the burned area corresponds to monocultures of exotic tree species (mainly pine and eucalyptus) (CONAF, 2019)*". (l. 661-663)

The same change is done in Appendix 1. (l.1839-1841)

Corporación Nacional Forestal (CONAF). 2019. Incendios forestales. Estadísticas históricas. http://www. conaf.cl/incendios-forestales/incendios-forestales-en- chile/estadisticas-historicas/

The other comment refers to a paragraph in the main text that summarizes some of the results presented in Appendix 1. We report the interrelation between different systems, where both local and global factors have a role.

- Firstly, the megadrought (precipitation deficits) and related streamflow deficits are directly related to climate change (global factor), and there are formal attribution exercises of this (e.g., Boisier et al., 2016, 2018).

- Secondly, there is a connection between the hydrological and the ecosystems. The available fire database from the last four decades indicates that the dry conditions during the megadrought are connected to the higher occurrence of wildfires and a larger burned area (González et al., 2018).

- Thirdly, the above connection is also modulated by the social system, which is inferred from the facts that almost half of the burned area corresponds to exotic tree plantations (anthropogenic perturbation). Also, a 99% of wildfires in Chile are initiated by human actions, which highlight the influence of local human perturbations on fires.

We modified the text to make this argument clearer (l.657-663).

2. In paragraph 700 I don't understand the phrase: "More research is needed on interactions and feedback between systems related to drought impacts and responses. For example, studies on the interactions between drought and wildfires should not only include ecological processes but also hydro-climatic and social processes". In my opinion, many studies have observed those cascading phenomena already as a prolonged drought changes social dynamics associated with crops, forestry, rural traditions, etc., and water scarcity makes them established long-term wildfire drivers. So, what more studies need to be made related to wildfires and drought? I suggest deleting it or reinforcing it if is relevant to the topic of interactions and feedback between systems.

>> We agree and deleted the example in the revised manuscript (l.774-775).

2. The appendix section provides abundant information about the cases but it is not homogeneous in the case selection (e.g. Chile presents few basins in the center, and the US presents one basin which is the Colorado River Basin (United States), The Horn of Africa (HOA) which is an entire region, etc.) and how they were extensively written. Also, I miss a global map in the beginning to locate the 5 cases. What happened with the ecosystem part of APPENDIX 3 - Case study Northeast Brazil? It is blank.

>> Thanks for these comments. Our aim was to retrieve existing evidence from different study cases. These studies were selected based on criteria including their geographical representation and providing evidence for several (but not all) sub-systems. Given this, the cases are in fact not

homogeneous, for example, the case of Chile includes the central region of the country, although some variables are presented as time series for specific locations. (Figure A.1).

For the Brazil case, in the revised version of Appendix 3, we have now added the ecosystem part:

*"The prolonged droughts experienced in Northeast Brazil, especially intense from 2011 to 2020, inflicted considerable damage on the Caatinga biome; such damages were exacerbated by land use and occupation practices (Caballero et al., 2023). The Caatinga is the only uniquely Brazilian biome, one of the world's most populated and biologically diverse semi-arid regions. However, it is considered to be one of the least studied biomes in Brazil despite undergoing significant changes in land use and cover, as well as facing unsustainable land resource utilization (Beuchle et al., 2015; Santos et al., 2011). The Caatinga semi-arid climate and heterogeneous vegetation cover consist of scrubland and seasonally dry forest (Leal et al., 2005; Santos et al., 2011). Human activities such as fires and deforestation have led to the loss of vegetation cover and increased soil water deficit, accelerating desertification. This initial desertification, compounded by intensified drought conditions, has furthered the desertification process, altered the microclimate, and hindered subsistence agriculture and rural development (Gutiérrez et al., 2014; Marengo et al., 2018; Silva et al., 2020; Tomasella et al., 2018). Consequently, the compromised resilience of the ecosystem increased vulnerability to future droughts and worsened socioeconomic conditions in the region."*

*Beuchle, R., Grecchi, R. C., Shimabukuro, Y. E., Seliger, R., Eva, H. D., Sano, E., & Achard, F. (2015). Land cover changes in the Brazilian Cerrado and Caatinga biomes from 1990 to 2010 based on a systematic remote sensing sampling approach. Applied Geography, 58, 116–127. https://doi.org/10.1016/j.apgeog.2015.01.017*

*Caballero, C. B., Biggs, T. W., Vergopolan, N., West, T. A. P., & Ruhoff, A. (2023). Transformation of Brazil's biomes: The dynamics and fate of agriculture and pasture expansion into native vegetation. Science of the Total Environment, 896. https://doi.org/10.1016/j.scitotenv.2023.166323*

*Gutiérrez, A. P. A., Engle, N. L., De Nys, E., Molejón, C., & Martins, E. S. (2014). Drought preparedness in Brazil. Weather and Climate Extremes, 3, 95–106. https://doi.org/10.1016/j.wace.2013.12.001*

*Leal, I. R., Maria Da Silva, J. C., Tabarelli, M., & Lacher Jr, T. E. (2005). Mudando o curso da conservação da biodiversidade na Caatinga do Nordeste do Brasil (Vol. 1).*

*Marengo, J. A., Alves, L. M., Alvala, R. C. S., Cunha, A. P., Brito, S., & Moraes, O. L. L. (2018). Climatic characteristics of the 2010-2016 drought in the semiarid northeast Brazil region. Anais Da Academia Brasileira de Ciencias, 90(2), 1973–1985. https://doi.org/10.1590/0001-3765201720170206*

*Santos, J. C., Leal, I. R., Almeida-Cortez, J. S., Fernandes, G. W., & Tabarelli, M. (2011). Caatinga: the scientific negligence experienced by a dry tropical forest. In Available online: www.tropicalconservationscience.org Mongabay.com Open Access Journal-Tropical Conservation Science (Vol. 4, Issue 3). www.tropicalconservationscience.org*

*Silva, J. L. B. da, Moura, G. B. de A., Silva, M. V. da, Lopes, P. M. O., Guedes, R. V. de S., Silva, Ê. F. de F. e., Ortiz, P. F. S., & Rodrigues, J. A. de M. (2020). Changes in the water resources, soil use and spatial dynamics of Caatinga vegetation cover over semiarid region of the Brazilian Northeast. Remote Sensing Applications: Society and Environment, 20. https://doi.org/10.1016/j.rsase.2020.100372*

*Tomasella, J., Silva Pinto Vieira, R. M., Barbosa, A. A., Rodriguez, D. A., de Oliveira Santana, M., & Sestini, M. F. (2018). Desertification trends in the Northeast of Brazil over the period 2000–2016. International Journal of Applied Earth Observation and Geoinformation, 73, 197–206. https://doi.org/10.1016/j.jag.2018.06.012*

---

## Author Response (AR2)

**Rebuttal (bold) to editor comment Floris van Ogtrop (normal text)**

Dear Dr. Anne van Loon and co-authors,

I'd like to thank you for providing an updated manuscript. Certainly an extensive, coordinated and thought provoking exploration of the complex impacts of drought on us and the (eco)systems that support us and a suitable highlight paper for the special issue Drought, society, and ecosystems.

I have contacted the referees and they agree that you have addressed their comments well and this is reflected in your most recent submission. Therefore, I am happy for this to go forward to publication. Please check the figure in the final case study which appears to be cropped at the top and bottom. Please also check position of brackets and grammar in the recently added text.

**>> Thanks for your positive evaluation of our manuscript. We have implemented all suggested changes. We included the uncropped figure for the Rhine case study and have addressed the brackets and grammar issues in the new text. We also updated two references which are now published.**

**Rebuttal (bold) to editor comment Kai Schröter (normal text)**

Dear Anne Van Loon and co-authors,
I agree with the editor's decision that only technical corrections are needed. In addition, please also check the font size in Figure 5. Some labels are not readable (e.g. nature-based solutions).
Best regards,
Kai Schröter

**>> Thanks for pointing out the issue with the font size in Figure 5. The font size of the text in Figure 5 has been increased from 6 points to 10 points.**